# BENEFITS OF DEPTH FOR LONG-TERM MEMORY OF RECURRENT NETWORKS

**Yoav Levine, Or Sharir & Amnon Shashua**
The Hebrew University of Jerusalem
{yoavlevine,or.sharir,shashua}@cs.huji.ac.il

## ABSTRACT

The key attribute that drives the unprecedented success of modern Recurrent Neural Networks (RNNs) on learning tasks which involve sequential data, is their ever-improving ability to model intricate long-term temporal dependencies. However, a well established measure of RNNs' long-term memory capacity is lacking, and thus formal understanding of their ability to correlate data throughout time is limited. Though depth efficiency in convolutional networks is well established by now, it does not suffice in order to account for the success of deep RNNs on inputs of varying lengths, and the need to address their 'time-series expressive power' arises. In this paper, we analyze the effect of depth on the ability of recurrent networks to express correlations ranging over long time-scales. To meet the above need, we introduce a measure of the information flow across time that can be supported by the network, referred to as the *Start-End separation rank*. Essentially, this measure reflects the distance of the function realized by the recurrent network from a function that models no interaction whatsoever between the beginning and end of the input sequence. We prove that deep recurrent networks support Start-End separation ranks which are exponentially higher than those supported by their shallow counterparts. Moreover, we show that the ability of deep recurrent networks to correlate different parts of the input sequence increases exponentially as the input sequence extends, while that of vanilla shallow recurrent networks does not adapt to the sequence length at all. Thus, we establish that depth brings forth an overwhelming advantage in the ability of recurrent networks to model long-term dependencies, and provide an exemplar of quantifying this key attribute which may be readily extended to other RNN architectures of interest, *e.g.* variants of LSTM networks. We obtain our results by considering a class of recurrent networks referred to as *Recurrent Arithmetic Circuits* (RACs), which merge the hidden state with the input via the Multiplicative Integration operation.

## 1 INTRODUCTION

Over the past few years, Recurrent Neural Networks (RNNs) have become the prominent machine learning architecture for modeling sequential data, having been successfully employed for language modeling (Sutskever et al., 2011; Graves, 2013), neural machine translation (Bahdanau et al., 2014), speech recognition (Graves et al., 2013; Amodei et al., 2016), and more. The success of recurrent networks in learning complex functional dependencies for sequences of varying lengths, readily implies that long-term and elaborate correlations in the given inputs are somehow supported by these networks. However, formal understanding of the influence of a recurrent network's structure on its expressiveness, and specifically on its ever-improving ability to integrate data throughout time (*e.g.* translating long sentences, answering elaborate questions), is lacking.

An ongoing empirical effort to successfully apply recurrent networks to tasks of increasing complexity and temporal extent, includes augmentations of the recurrent unit such as Long Short Term Memory (LSTM) networks (Hochreiter and Schmidhuber, 1997) and their variants (*e.g.* Cho et al. (2014)). A parallel avenue, which we focus on in this paper, includes the stacking of layers to form deep recurrent networks (Schmidhuber, 1992). Deep recurrent networks, which exhibit empirical superiority over shallow ones (see *e.g.* Graves et al. (2013)), implement hierarchical processing of information at every time-step that accompanies their inherent time-advancing computation. Evidence for a time-scale related effect arises from experiments (Hermans and Schrauwen, 2013) – deep

recurrent networks appear to model correlations which correspond to longer time-scales than shallow ones. These findings, which imply that depth brings forth a considerable advantage in complexity and in temporal capacity of recurrent networks, have no adequate theoretical explanation.

In this paper, we address the above presented issues. Based on the relative maturity of *depth efficiency* results in neural networks, namely results that show that deep networks efficiently express functions that would require shallow ones to have a super-polynomial size (*e.g.* Cohen et al. (2016); Eldan and Shamir (2016)), it is natural to assume that depth has a similar effect on the expressiveness of recurrent networks. Indeed, we show that depth efficiency holds for recurrent networks.

However, the distinguishing attribute of recurrent networks, is their inherent ability to cope with varying input sequence length. Thus, once establishing the above depth efficiency in recurrent networks, a basic question arises, which relates to the apparent depth enhanced long-term memory in recurrent networks: *Do the functions which are efficiently expressed by deep recurrent networks correspond to dependencies over longer time-scales?* We answer this question, by showing that depth provides an exponential boost to the ability of recurrent networks to model long-term dependencies.

In order to take-on the above question, we introduce in section 2 a recurrent network referred to as a *recurrent arithmetic circuit* (RAC) that shares the architectural features of RNNs, and differs from them in the type of non-linearity used in the calculation. This type of connection between state-of-the-art machine learning algorithms and arithmetic circuits (also known as Sum-Product Networks (Poon and Domingos, 2011)) has well-established precedence in the context of neural networks. Delalleau and Bengio (2011) prove a depth efficiency result on such networks, and Cohen et al. (2016) theoretically analyze the class of Convolutional Arithmetic Circuits which differ from common ConvNets in the exact same fashion in which RACs differ from more standard RNNs. Conclusions drawn from such analyses were empirically shown to extend to common ConvNets (*e.g.* Sharir and Shashua (2017); Levine et al. (2017)). Beyond their connection to theoretical models, the modification which defines RACs resembles that of Multiplicative RNNs (Sutskever et al., 2011) and of Multiplicative Integration networks (Wu et al., 2016), which provide a substantial performance boost over many of the existing RNN models. In order to obtain our results, we make a connection between RACs and the Tensor Train (TT) decomposition (Oseledets, 2011), which suggests that Multiplicative RNNs may be related to a generalized TT-decomposition, similar to the way Cohen and Shashua (2016) connected ReLU ConvNets to generalized tensor decompositions.

We move on to introduce in section 3 the notion of *Start-End separation rank* as a measure of the recurrent network's ability to model elaborate long-term dependencies. In order to analyze the long-term correlations of a function over a sequential input which extends $T$ time-steps, we partition the inputs to those which arrive at the first $T/2$ time-steps ("Start") and the last $T/2$ time-steps ("End"), and ask how far the function realized by the recurrent network is from being separable w.r.t. this partition. Distance from separability is measured through the notion of separation rank (Beylkin and Mohlenkamp, 2002), which can be viewed as a surrogate of the $L^2$ distance from the closest separable function. For a given function, high Start-End separation rank implies that the function induces strong correlation between the beginning and end of the input sequence, and vice versa.

In section 4 we directly address the depth enhanced long-term memory question above, by examining depth $L = 2$ RACs and proving that functions realized by these deep networks enjoy Start-End separation ranks that are exponentially higher than those of shallow networks, implying that indeed these functions can model more elaborate input dependencies over longer periods of time. An additional reinforcing result is that the Start-End separation rank of the deep recurrent network grows exponentially with the sequence length, while that of the shallow recurrent network is *independent* of the sequence length. Informally, this implies that vanilla shallow recurrent networks are inadequate in modeling correlations of long input sequences, since in contrast to the case of deep recurrent networks, the modeled dependencies achievable by shallow ones do not adapt to the actual length of the input. Finally, we present and motivate a quantitative conjecture by which the Start-End separation rank of recurrent networks grows exponentially with the network depth. A proof of this conjecture, which will provide an even deeper insight regarding the advantages of depth in recurrent networks, is left as an open problem.

## 2 RECURRENT ARITHMETIC CIRCUITS

In this section, we introduce a class of recurrent networks referred to as Recurrent Arithmetic Circuits (RACs), which shares the architectural features of standard RNNs. As demonstrated below,

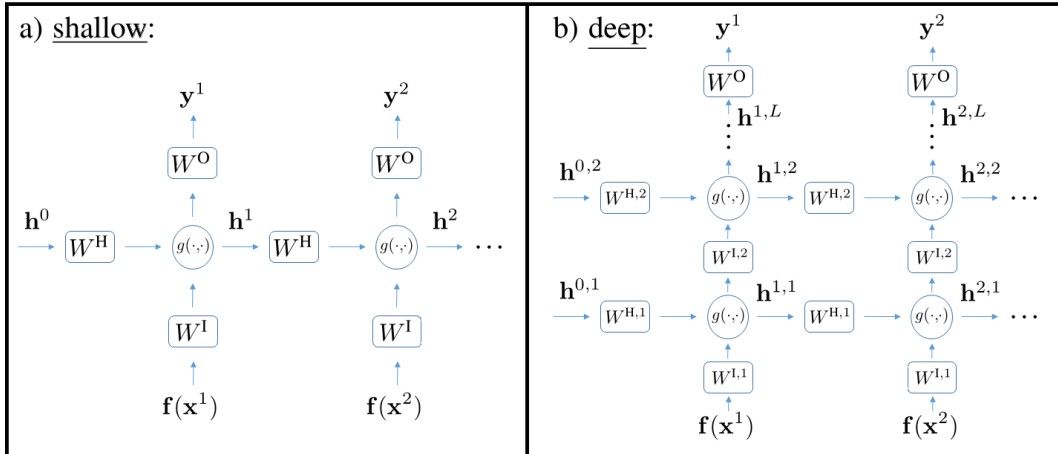

Figure 1: Shallow and deep recurrent networks, as described by eqs. 1 and 4, respectively.

the operation of RACs on sequential data is identical to the operation of RNNs, where a hidden state mixes information from previous time-steps with new incoming data (see fig. 1). The two classes differ only in the type of non-linearity used in the calculation, as described by eqs. 1-3. In the following sections, we utilize the algebraic properties of RACs for proving results regarding their ability to model long-term dependencies of their inputs.

We present below the basic framework of shallow recurrent networks (fig. 1(a)), which describes both the common RNNs and the newly introduced RACs. A recurrent network is a network that models a discrete-time dynamical system; we focus on an example of a sequence to sequence classification task into one of the categories $\{1, ..., C\} \equiv [C]$. Denoting the temporal dependence by $t$, the sequential input to the network is $\{\mathbf{x}^t \in \mathcal{X}\}_{t=1}^T$, and the output is a sequence of class scores vectors $\{\mathbf{y}^{t,L,\Theta} \in \mathbb{R}^C\}_{t=1}^T$, where $L$ is the network depth, $\Theta$ denotes the parameters of the recurrent network, and $T$ represents the extent of the sequence in time-steps. We assume the input lies in some input space $\mathcal{X}$ that may be discrete (e.g. text data) or continuous (e.g. audio data), and that some initial mapping $\mathbf{f} : \mathcal{X} \to \mathbb{R}^M$ is preformed on the input, so that all input types are mapped to vectors $\mathbf{f}(\mathbf{x}^t) \in \mathbb{R}^M$. The function $\mathbf{f}(\cdot)$ may be viewed as an encoding, e.g. words to vectors or images to a final dense layer via some trained ConvNet. The output at time $t \in [T]$ of the shallow (depth $L = 1$) recurrent network with $R$ hidden channels, depicted in fig. 1(a), is given by:

$$\mathbf{h}^t = g\left(W^\mathrm{H}\mathbf{h}^{t-1}, W^\mathrm{I}\mathbf{f}(\mathbf{x}^t)\right) \tag{1}$$
$$\mathbf{y}^{t,1,\Theta} = W^\mathrm{O}\mathbf{h}^t,$$

where $\mathbf{h}^t \in \mathbb{R}^R$ is the hidden state of the network at time $t$ ($\mathbf{h}^0$ is some initial hidden state), $\Theta$ denotes the learned parameters $W^\mathrm{I} \in \mathbb{R}^{R \times M}, W^\mathrm{H} \in \mathbb{R}^{R \times R}, W^\mathrm{O} \in \mathbb{R}^{C \times R}$, which are the input, hidden and output weights matrices respectively, and $g$ is some non-linear operation. A bias term is usually added to eq. 1, however, because it bears no effect on our analysis, we omit it for simplicity. For common RNNs, the non-linearity is given by:

$$g^\mathrm{RNN}(\mathbf{a}, \mathbf{b}) = \sigma(\mathbf{a} + \mathbf{b}), \tag{2}$$

where $\sigma(\cdot)$ is typically some point-wise non-linearity such as sigmoid, tanh etc. For the newly introduced class of RACs, $g$ is given by:

$$g^\mathrm{RAC}(\mathbf{a}, \mathbf{b}) = \mathbf{a} \odot \mathbf{b}, \tag{3}$$

where the operation $\odot$ stands for element-wise multiplication between vectors, for which the resultant vector upholds $(\mathbf{a} \odot \mathbf{b})_i = a_i \cdot b_i$. This form of merging the input and the hidden state by multiplication rather than addition is referred to as Multiplicative Integration (Wu et al., 2016).

The extension to deep recurrent networks is natural, and we follow the common approach (see e.g. Hermans and Schrauwen (2013)) where each layer acts as a recurrent network which receives the hidden state of the previous layer as its input. The output at time $t$ of the depth $L$ recurrent network

with $R$ hidden channels in each layer,[1] depicted in fig. 1(b), is constructed by the following:

$$\mathbf{h}^{t,l} = g\left(W^{\mathrm{H},l}\mathbf{h}^{t-1,l}, W^{\mathrm{I},l}\mathbf{h}^{t,l-1}\right)$$
$$\mathbf{h}^{t,0} \equiv \mathbf{f}(\mathbf{x}^t) \tag{4}$$
$$\mathbf{y}^{t,L,\Theta} = W^{\mathrm{O}}\mathbf{h}^{t,L},$$

where $\mathbf{h}^{t,l} \in \mathbb{R}^R$ is the state of the depth $l$ hidden unit at time $t$ ($\mathbf{h}^{0,l}$ is some initial hidden state per layer), and $\Theta$ denotes the learned parameters. Specifically, $W^{\mathrm{I},l} \in \mathbb{R}^{R \times R}$ ($l > 1$), $W^{\mathrm{H},l} \in \mathbb{R}^{R \times R}$ are the input and hidden weights matrices at depth $l$, respectively. For $l = 1$, the weights matrix which multiplies the inputs vector has the appropriate dimensions: $W^{\mathrm{I},1} \in \mathbb{R}^{R \times M}$. The output weights matrix is $W^{\mathrm{O}} \in \mathbb{R}^{C \times R}$ as in the shallow case, representing a final calculation of the scores for all classes 1 through $C$ at every time-step. The non-linear operation $g$ determines the type of the deep recurrent network, where a common deep RNN is obtained by choosing $g = g^{\mathrm{RNN}}$ (eq. 2), and a deep RAC is obtained for $g = g^{\mathrm{RAC}}$ (eq. 3).

We consider the newly presented class of RACs to be a good surrogate of common RNNs. Firstly, there is an obvious structural resemblance between the two classes, as the recurrent aspect of the calculation has the exact same form in both networks (fig. 1). In fact, recurrent networks that include Multiplicative Integration similarly to RACs, have been shown to outperform many of the existing RNN models (Sutskever et al., 2011; Wu et al., 2016). Secondly, as mentioned above, arithmetic circuits have been successfully used as surrogates of convolutional networks. The fact that Cohen and Shashua (2016) laid the foundation for extending the proof methodologies of convolutional arithmetic circuits to common ConvNets with ReLU activations, suggests that such adaptations may be made in the recurrent network analog, rendering the newly proposed class of recurrent networks all the more interesting. In the following sections, we make use of the algebraic properties of RACs in order to obtain clear-cut observations regarding the benefits of depth in recurrent networks.

## 3    TEMPORAL CORRELATIONS MODELED BY RECURRENT NETWORKS

In this section, we establish means for quantifying the ability of recurrent networks to model long-term temporal dependencies in the sequential input data. We begin by introducing the Start-End separation-rank of the function realized by a recurrent network as a measure of the amount of information flow across time that can be supported by the network. We then tie the Start-End separation rank to the algebraic concept of grid tensors (Hackbusch, 2012), which will allow us to employ tools and results from tensorial analysis in order to show that depth provides an exponential boost to the ability of recurrent networks to model elaborate long-term temporal dependencies.

### 3.1    THE "START-END" SEPARATION RANK

We define below the concept of the *Start-End separation rank* for functions realized by recurrent networks after $T$ time-steps, *i.e.* real functions that take as input $X = (\mathbf{x}^1, \dots, \mathbf{x}^T) \in \mathcal{X}^T$. The separation rank quantifies a function's distance from separability with respect to two disjoint subsets of its inputs. Specifically, let $(S, E)$ be a partition of input indices, such that $S = \{1, \dots, T/2\}$ and $E = \{T/2 + 1, \dots, T\}$ (we consider even values of $T$ throughout the paper for convenience of presentation). This implies that $\{\mathbf{x}^s\}_{s \in S}$ are the first $T/2$ ("Start") inputs to the network, and $\{\mathbf{x}^e\}_{e \in E}$ are the last $T/2$ ("End") inputs to the network. For a function $y : \mathcal{X}^T \to \mathbb{R}$, the *Start-End separation rank* is defined as follows:

$$\mathrm{sep}_{(S,E)}(y) \equiv \min\left\{K \in \mathbb{N} \cup \{0\} : \exists g_1^s \dots g_K^s : \mathcal{X}^{T/2} \to \mathbb{R}, \ g_1^e \dots g_K^e : \mathcal{X}^{T/2} \to \mathbb{R} \ s.t. \tag{5}\right.$$

$$\left. y(\mathbf{x}^1, \dots, \mathbf{x}^T) = \sum_{\nu=1}^{K} g_\nu^s(\mathbf{x}^1, \dots, \mathbf{x}^{T/2}) g_\nu^e(\mathbf{x}^{T/2+1}, \dots, \mathbf{x}^T)\right\}.$$

In words, it is the minimal number of summands that together give $y$, where each summand is *separable w.r.t.* $(S, E)$, *i.e.* is equal to a product of two functions – one that intakes only inputs from the first $T/2$ time-steps, and another that intakes only inputs from the last $T/2$ time-steps.

The separation rank w.r.t. a general partition of the inputs was introduced in Beylkin and Mohlenkamp (2002) for high-dimensional numerical analysis, and was employed for various applications, *e.g.* chemistry (Harrison et al., 2003), particle engineering (Hackbusch, 2006), and machine learning (Beylkin et al., 2009). Cohen and Shashua (2017) connect the separation rank to the

---

[1]We assume for simplicity that the number of hidden channels is the same for all layers. See Levine et al. (2017) for treatment of the channel numbers' effect on the expressivity of convolutional networks .

$L^2$ distance of the function from the set of separable functions, and use it to measure correlations modeled by deep convolutional networks. Levine et al. (2017) tie the separation rank to the family of quantum entanglement measures, which quantify correlations in many-body quantum systems.

In our context, if the Start-End separation rank of a function realized by a recurrent network is equal to 1, then the function is separable, meaning it cannot model any interaction between the inputs which arrive at the beginning of the sequence and the inputs that follow later, towards the end of the sequence. Specifically, if $\text{sep}_{(S,E)}(y) = 1$ then there exist $g^s : \mathcal{X}^{T/2} \to \mathbb{R}$ and $g^e : \mathcal{X}^{T/2} \to \mathbb{R}$ such that $y(\mathbf{x}^1, \ldots, \mathbf{x}^T) = g^s(\mathbf{x}^1, \ldots, \mathbf{x}^{T/2}) g^e(\mathbf{x}^{T/2+1}, \ldots, \mathbf{x}^T)$, and the function $y$ cannot take into account consistency between the values of $\{\mathbf{x}^1, \ldots, \mathbf{x}^{T/2}\}$ and those of $\{\mathbf{x}^{T/2+1}, \ldots, \mathbf{x}^T\}$. In a statistical setting, if $y$ were a probability density function, this would imply that $\{\mathbf{x}^1, \ldots, \mathbf{x}^{T/2}\}$ and $\{\mathbf{x}^{T/2+1}, \ldots, \mathbf{x}^T\}$ are statistically independent. The higher $\text{sep}_{(S,E)}(y)$ is, the farther $y$ is from this situation, *i.e.* the more it models dependency between the beginning and the end of the inputs sequence. Stated differently, if the recurrent network's architecture restricts the hypothesis space to functions with low Start-End separation ranks, a more elaborate long-term temporal dependence, which corresponds to a function with a higher Start-End separation rank, cannot be learned.

In section 4 we show that deep RACs support Start-End separations ranks which are exponentially larger than those supported by shallow RACs, and are therefore much better fit to model long-term temporal dependencies. To this end, we employ in the following sub-section the algebraic tool of *grid tensors* that will allow us to evaluate the Start-End separation ranks of deep and shallow RACs.

### 3.2 Bounding the Start-End Separation Rank via Grid Tensors

We begin by laying out basic concepts in tensor theory required for the upcoming analysis. The core concept of a *tensor* may be thought of as a multi-dimensional array. The *order* of a tensor is defined to be the number of indexing entries in the array, referred to as *modes*. The *dimension* of a tensor in a particular mode is defined as the number of values taken by the index in that mode. If $\mathcal{A}$ is a tensor of order $T$ and dimension $M_i$ in each mode $i \in [T]$, its entries are denoted $\mathcal{A}_{d_1 \ldots d_T}$, where the index in each mode takes values $d_i \in [M_i]$. A fundamental operator in tensor analysis is the *tensor product*, which we denote by $\otimes$. It is an operator that intakes two tensors $\mathcal{A} \in \mathbb{R}^{M_1 \times \cdots \times M_P}$ and $\mathcal{B} \in \mathbb{R}^{M_{P+1} \times \cdots \times M_{P+Q}}$, and returns a tensor $\mathcal{A} \otimes \mathcal{B} \in \mathbb{R}^{M_1 \times \cdots \times M_{P+Q}}$ defined by: $(\mathcal{A} \otimes \mathcal{B})_{d_1 \ldots d_{P+Q}} = \mathcal{A}_{d_1 \ldots d_P} \cdot \mathcal{B}_{d_{P+1} \ldots d_{P+Q}}$. An additional concept we will make use of is the *matricization of $\mathcal{A}$ w.r.t. the partition $(S, E)$*, denoted $[\![\mathcal{A}]\!]_{S,E}$, which is essentially the arrangement of the tensor elements as a matrix whose rows correspond to $S$ and columns to $E$ (formally presented in appendix C).

We consider the function realized by a shallow RAC with $R$ hidden channels, which computes the score of class $c \in [C]$ at time $T$. This function, which is given by a recursive definition in eqs. 1 and 3, can be alternatively written in the following closed form:

$$y_c^{T,1,\Theta}(\mathbf{x}^1, \ldots, \mathbf{x}^T) = \sum_{d_1 \ldots d_T = 1}^M \left(\mathcal{A}_c^{T,1,\Theta}\right)_{d_1, \ldots, d_T} \prod_{i=1}^T f_{d_i}(\mathbf{x}^i), \tag{6}$$

where the order $T$ tensor $\mathcal{A}_c^{T,1,\Theta}$, which lies at the heart of the above expression, is referred to as the *shallow RAC weights tensor*, since its entries are polynomials in the network weights $\Theta$. Specifically, denoting the rows of the input weights matrix, $W^{\text{I}}$, by $\mathbf{a}^{\text{I},\alpha} \in \mathbb{R}^M$ (or element-wise: $a_j^{\text{I},\alpha} = W_{\alpha,j}^{\text{I}}$), the rows of the hidden weights matrix, $W^{\text{H}}$, by $\mathbf{a}^{\text{H},\beta} \in \mathbb{R}^R$ (or element-wise: $a_j^{\text{H},\beta} = W_{\beta,j}^{\text{H}}$), and the rows of the output weights matrix, $W^{\text{O}}$, by $\mathbf{a}^{\text{O},c} \in \mathbb{R}^R$, $c \in [C]$ (or element-wise: $a_j^{\text{O},c} = W_{c,j}^{\text{O}}$), the shallow RAC weights tensor can be gradually constructed in the following fashion:

$$\underbrace{\phi^{2,\beta}}_{\text{order 2 tensor}} = \sum_{\alpha=1}^R a_\alpha^{\text{H},\beta} \, \mathbf{a}^{\text{I},\alpha} \otimes \mathbf{a}^{\text{I},\alpha}$$

$$\cdots$$

$$\underbrace{\phi^{t,\beta}}_{\text{order } t \text{ tensor}} = \sum_{\alpha=1}^R a_\alpha^{\text{H},\beta} \phi^{t-1,\alpha} \otimes \mathbf{a}^{\text{I},\alpha}$$

$$\cdots$$

$$\underbrace{\mathcal{A}_c^{T,1,\Theta}}_{\text{order } T \text{ tensor}} = \sum_{\alpha=1}^R a_\alpha^{\text{O},c} \phi^{T-1,\alpha} \otimes \mathbf{a}^{\text{I},\alpha}, \tag{7}$$

having set $\mathbf{h}^0 = \left(W^{\text{H}}\right)^\dagger \mathbf{1}$, where $\dagger$ is the pseudoinverse operation. In the above equation, the tensor products, which appear inside the sums, are directly related to the Multiplicative Integration property

of RACs (eq. 3). The sums originate in the multiplication of the hidden states vector by the hidden weights matrix at every time-step (eq. 1). The construction of the shallow RAC weights tensor, presented in eq. 7, is referred to as a Tensor Train (TT) decomposition of TT-rank $R$ in the tensor analysis community and is analogously described by a Matrix Product State (MPS) Tensor Network (see Orús (2014)) in the quantum physics community. See appendix A for the Tensor Networks construction of deep and shallow RACs, which provides graphical insight regarding the exponential complexity brought forth by depth in recurrent networks.

We now present the concept of grid tensors, which are a form of function discretization. Essentially, the function is evaluated for a set of points on an exponentially large grid in the input space and the outcomes are stored in a tensor. Formally, fixing a set of *template* vectors $\mathbf{x}^{(1)}, \ldots, \mathbf{x}^{(M)} \in \mathcal{X}$, the points on the grid are the set $\{(\mathbf{x}^{(d_1)}, \ldots, \mathbf{x}^{(d_T)})\}_{d_1, \ldots, d_T=1}^M$. Given a function $y(\mathbf{x}^1, \ldots, \mathbf{x}^T)$, the set of its values on the grid arranged in the form of a tensor are called the grid tensor induced by $y$, denoted $\mathcal{A}(y)_{d_1, \ldots, d_T} \equiv y(\mathbf{x}^{(d_1)}, \ldots, \mathbf{x}^{(d_T)})$. The grid tensors of functions realized by recurrent networks, will allow us to calculate their separations ranks and establish definitive conclusions regarding the benefits of depth these networks. Having presented the tensorial structure of the function realized by a shallow RAC, as given by eqs. 6 and 7 above, we are now in a position to tie its Start-End separation rank to its grid tensor, as formulated in the following claim:

**Claim 1.** *Let $y_c^{T,1,\Theta}$ be a function realized by a shallow RAC (fig. 1(a)) after $T$ time-steps, and let $\mathcal{A}_c^{T,1,\Theta}$ be its shallow RAC weights tensor, constructed according to eq. 7. Assume that the network's initial mapping functions $\{f_d\}_{d=1}^M$ are linearly independent, and that they, as well as the functions $g_\nu, g_\nu'$ in the definition of Start-End separation rank (eq. 5), are measurable and square-integrable.[2] Then, there exist template vectors $\mathbf{x}^{(1)}, \ldots, \mathbf{x}^{(M)} \in \mathcal{X}$ such that the following holds:*

$$\text{sep}_{(S,E)}\left(y_c^{T,1,\Theta}\right) = \text{rank}\left(\llbracket \mathcal{A}(y_c^{T,1,\Theta}) \rrbracket_{S,E}\right) = \text{rank}\left(\llbracket \mathcal{A}_c^{T,1,\Theta} \rrbracket\right)_{S,E}, \tag{8}$$

*where $\mathcal{A}(y_c^{T,1,\Theta})$ is the grid tensor of $y_c^{T,1,\Theta}$ with respect to the above template vectors.*

*Proof.* See appendix B.1. □

The above claim establishes an equality between the Start-End separation rank and the rank of the matrix obtained by the corresponding grid tensor matricization, denoted $\llbracket \mathcal{A}(y_c^{T,1,\Theta}) \rrbracket_{S,E}$, with respect to a specific set of template vectors. Note that the limitation to specific template vectors does not restrict our results, as grid tensors are merely a tool used to bound the separation rank. The additional equality to the rank of the matrix obtained by matricizing the shallow RAC weights tensor, will be of use to us when proving our main results below (theorem 1).

Due to the inherent use of data duplication in the computation preformed by a deep RAC (see appendix A.3 for further details), it cannot be written in a closed tensorial form similar to that of eq. 6. This in turn implies that the equality shown in claim 1 does not hold for functions realized by deep RACs. The following claim introduces a fundamental relation between a function's Start-End separation rank and the rank of the matrix obtained by the corresponding matricization. This relation, which holds for all functions, is formulated below for functions realized by deep RACs:

**Claim 2.** *Let $y_c^{T,L,\Theta}$ be a function realized by a depth $L$ RAC (fig. 1(b)) after $T$ time-steps. Then, for any template vectors $\mathbf{x}^{(1)}, \ldots, \mathbf{x}^{(M)} \in \mathcal{X}$ it holds that:*

$$\text{sep}_{(S,E)}\left(y_c^{T,L,\Theta}\right) \geq \text{rank}\left(\llbracket \mathcal{A}(y_c^{T,L,\Theta}) \rrbracket_{S,E}\right), \tag{9}$$

*where $\mathcal{A}(y_c^{T,L,\Theta})$ is the grid tensor of $y_c^{T,L,\Theta}$ with respect to the above template vectors.*

*Proof.* See appendix B.2. □

Claim 2 will allow us to provide a lower bound on the Start-End separation rank of functions realized by deep RACs, which we show to be exponentially higher than the Start-End separation rank of functions realized by shallow RACs (to be obtained via claim 1). Thus, in the next section, we employ the above presented tools to show that an exponential enhancement of the Start-End separation rank is brought forth by depth in recurrent networks.

---

[2] Square-integrability may seem as a limitation at first glance, as for example neurons $f_d(\mathbf{x}) = \sigma(\mathbf{w}_d^\top \mathbf{x} + b_d)$ with sigmoid or ReLU activation $\sigma(\cdot)$, do not meet this condition. However, since in practice our inputs are bounded (*e.g.* image pixels by holding intensity values, etc), we may view functions as having compact support, which, as long as they are continuous (holds in all cases of interest), ensures square-integrability.

# 4    DEPTH ENHANCED LONG-TERM MEMORY IN RECURRENT NETWORKS

In this section, we present the main theoretical contributions of this paper. In section 4.1, we formally present a result which exponentially separates between the memory capacity of a deep ($L = 2$) recurrent network and a shallow ($L = 1$) one. Following the formal presentation of results in theorem 1, we discuss some of their implications and then conclude by sketching a proof outline for the theorem (full proof is relegated to appendix B.3). In section 4.2, we present a quantitative conjecture regarding the enhanced memory capacity of deep recurrent networks of general depth $L$, which relies on the inherent combinatorial properties of the recurrent network's computation. We leave the formal proof of this conjecture for future work.

## 4.1    SEPARATING BETWEEN SHALLOW AND DEEP RECURRENT NETWORKS

Theorem 1 states, that the correlations modeled between the beginning and end of the input sequence to a recurrent network, as measured by the Start-End separation rank (see section 3.1), can be exponentially more complex for deep networks than for shallow ones:

**Theorem 1.** *Let $y_c^{T,L,\Theta}$ be the function computing the output after $T$ time-steps of an RAC with $L$ layers, $R$ hidden channels per layer, weights denoted by $\Theta$, and initial hidden states $\mathbf{h}^{0,l}$, $l \in [L]$ (fig. 1 with $g = g^{\mathrm{RAC}}$). Assume that the network's initial mapping functions $\{f_d\}_{d=1}^M$ are linearly independent. Let $\mathrm{sep}_{(S,E)}\left(y_c^{T,L,\Theta}\right)$ be the Start-End separation rank of $y_c^{T,L,\Theta}$ (eq. 5). Then, the following holds almost everywhere,* i.e. *for all values of $\Theta \times \mathbf{h}^{0,l}$ but a set of Lebesgue measure zero:*

*1. $\mathrm{sep}_{(S,E)}\left(y_c^{T,L,\Theta}\right) = \min\left\{R, M^{T/2}\right\}$, for $L = 1$ (shallow network).*

*2. $\mathrm{sep}_{(S,E)}\left(y_c^{T,L,\Theta}\right) \geq \min\left\{\left(\!\!\binom{\min\{M,R\}}{T/2}\!\!\right), M^{T/2}\right\}$, for $L = 2$ (deep network),*

*where $\left(\!\!\binom{\min\{M,R\}}{T/2}\!\!\right)$ is the multiset coefficient, given in the binomial form by $\binom{\min\{M,R\}+T/2-1}{T/2}$.*

The above theorem readily implies that depth entails an enhanced ability of recurrent networks to model long-term temporal dependencies in the sequential input. Specifically, theorem 1 indicates depth efficiency – it ensures us that upon randomizing the weights of a deep RAC with $R$ hidden channels per layer, with probability 1 the function realized by it after $T$ time-steps may only be realized by a shallow RAC with a number of hidden channels that is exponentially large.[3] Stated alternatively, this means that almost all functional dependencies which lie in the hypothesis space of deep RACs with $R$ hidden channels per layer, calculated after $T$ time-steps, are inaccessible to shallow RACs with less than an exponential number of hidden channels. Thus, a shallow recurrent network would require exponentially more parameters than a deep recurrent network, if it is to implement the same function.

The established role of the Start-End separation rank as a correlation measure between the beginning and the end of the sequence (see section 3.1), implies that these functions, which are realized by almost any deep network and can never be realized by a shallow network of a reasonable size, represent more elaborate correlations over longer periods of time. The above notion is strengthened by the fact that the Start-End separation rank of deep RACs increases with the sequence length $T$, while the Start-End separation rank of shallow RACs is independent of it. This indicates that shallow recurrent networks are much more restricted in modeling long-term correlations than the deep ones, which enjoy an exponentially increasing Start-End separation rank as time progresses. Below, we present an outline of the proof for theorem 1 (see appendix B.3 for the full version):

*Proof sketch of theorem 1.*

1. For a shallow network, claim 1 establishes that the Start-End separation rank of the function realized by a shallow ($L = 1$) RAC is equal to the rank of the matrix obtained by matricizing the corresponding shallow RAC weights tensor (eq. 6) according to the Start-End partition: $\mathrm{sep}_{(S,E)}\left(y_c^{T,1,\Theta}\right) = \mathrm{rank}\left(\llbracket\mathcal{A}_c^{T,1,\Theta}\rrbracket\right)_{S,E}$. Thus, it suffices to prove that $\mathrm{rank}\left(\llbracket\mathcal{A}_c^{T,1,\Theta}\rrbracket\right)_{S,E} = R$ in order to satisfy bullet (1) of the theorem, as the rank is trivially upper-bounded by the dimension of the matrix, $M^{T/2}$. To this end, we call upon the TT-decomposition of $\mathcal{A}_c^{T,1,\Theta}$, given by eq. 7, which corresponds to the MPS Tensor Network presented in appendix A. We rely on a recent result by Levine et al. (2017), who

---

[3]The combinatorial coefficient $\left(\!\!\binom{\min\{M,R\}}{T/2}\!\!\right) = \binom{\min\{M,R\}+T/2-1}{T/2}$ is exponentially dependent on $\bar{R} \equiv \min\{M,R\}$: for $T > 2*(\bar{R}-1)$ this value is larger than $\frac{4^{\bar{R}}}{4\sqrt{\pi(\bar{R}-1)}}$ (for sufficiently large values of $\bar{R}$).

state that the rank of the matrix obtained by matricizing any tensor according to a partition $(S, E)$, is equal to a min-cut separating $S$ from $E$ in the Tensor Network graph representing this tensor. The required equality follows from the fact that the TT-decomposition in eq. 7 is of TT-rank $R$, which in turn implies that the min-cut in the appropriate Tensor Network graph is equal to $R$.

2. For a deep network, claim 2 assures us that the Start-End separation rank of the function realized by a depth $L = 2$ RAC is lower bounded by the rank of the matrix obtained by the corresponding grid tensor matricization: $\text{sep}_{(S,E)}\left(y_c^{T,L,\Theta}\right) \geq \text{rank}\left(\llbracket \mathcal{A}(y_c^{T,L,\Theta}) \rrbracket_{S,E}\right)$.

Thus, proving that $\text{rank}\left(\llbracket \mathcal{A}(y_c^{T,L,\Theta}) \rrbracket_{S,E}\right) \geq \left(\!\!\left(\begin{array}{c} \min\{M,R\} \\ T/2 \end{array}\right)\!\!\right)$ for all of the values of parameters $\Theta \times \mathbf{h}^{0,l}$ but a set of Lebesgue measure zero, would satisfy the theorem, and again, the rank is trivially upper-bounded by the dimension of the matrix, $M^{T/2}$. We use a lemma proved in Sharir et al. (2016), which states that since the entries of $\mathcal{A}(y_c^{T,L,\Theta})$ are polynomials in the deep recurrent network's weights, it suffices to find a single example for which the rank of the matricized grid tensor is greater than the desired lower bound. Finding such an example would indeed imply that for almost all of the values of the network parameters, the desired inequality holds. We choose a weight assignment such that the resulting matricized grid tensor resembles a matrix obtained by raising a rank-$\bar{R} \equiv \min\{M, R\}$ matrix to the Hadamard power of degree $T/2$. This operation, which raises each element of the original rank-$\bar{R}$ matrix to the power of $T/2$, was shown to yield a matrix with a rank upper-bounded by the multiset coefficient $\left(\!\!\left(\begin{array}{c} \bar{R} \\ T/2 \end{array}\right)\!\!\right)$ (see *e.g.* Amini et al. (2012)). We show that our assignment results in a matricized grid tensor with a rank which is not only upper-bounded by this value, but actually achieves it.

$\square$

## 4.2 INCREASE OF MEMORY CAPACITY WITH DEPTH

Theorem 1 provides a lower bound of $\left(\!\!\left(\begin{array}{c} R \\ T/2 \end{array}\right)\!\!\right)$ on the Start-End separation rank of depth $L = 2$ recurrent networks, exponentially separating deep recurrent networks from shallow ones. By a trivial assignment of weights in higher layers, the Start-End separation rank of even deeper recurrent networks ($L > 2$) is also lower-bounded by this expression, which does not depend on $L$. In the following, we conjecture that a tighter lower bound holds for networks of depth $L > 2$, the form of which implies that the memory capacity of deep recurrent networks grows exponentially with the network depth:

**Conjecture 1.** *Under the same conditions as in theorem 1, for all values of $\Theta \times \mathbf{h}^{0,l}$ but a set of Lebesgue measure zero, it holds for any $L$ that:*

$$\text{sep}_{(S,E)}\left(y_c^{T,L,\Theta}\right) \geq \min\left\{ \left(\!\!\left(\begin{array}{c} \min\{M,R\} \\ \left(\!\left(\begin{array}{c} T/2 \\ L-1 \end{array}\right)\!\right) \end{array}\right)\!\!\right), M^{T/2} \right\}.$$

We motivate conjecture 1 by investigating the combinatorial nature of the computation performed by a deep RAC. By constructing Tensor Networks which correspond to deep RACs, we attain an informative visualization of this combinatorial perspective. In appendix A, we provide full details of this construction and present the formal motivation for the conjecture. Below, we qualitatively outline this combinatorial approach.

A Tensor Network is essentially a graphical tool for representing algebraic operations which resemble multiplications of vectors and matrices, between higher order tensors. Fig. 2 shows an example of the Tensor Network representing the computation of a depth $L = 3$ RAC after $T = 6$ time-steps. This well-defined computation graph hosts the values of the weight matrices at its nodes. The inputs $\{x^1, \ldots, x^T\}$ are marked by their corresponding time-step $\{1, \ldots, T\}$, and are integrated in a depth dependent and time-advancing manner (see further discussion regarding this form in appendix A.3), as portrayed in the example of fig. 2. We highlight in red the basic unit in the Tensor Network which connects "Start" inputs $\{1, \ldots, T/2\}$ and "End" inputs $\{T/2+1, \ldots, T\}$. In order to estimate a lower bound on the Start-End separation rank of a depth $L > 2$ recurrent network, we employ a similar strategy to that presented in the proof sketch of the $L = 2$ case (see section 4.1). Specifically, we rely on the fact that it is sufficient to find a specific instance of the network parameters $\Theta \times \mathbf{h}^{0,l}$ for which $\llbracket \mathcal{A}(y_c^{T,L,\Theta}) \rrbracket_{S,E}$ achieves a certain rank, in order for this rank to bound the Start-End separation rank of the network from below.

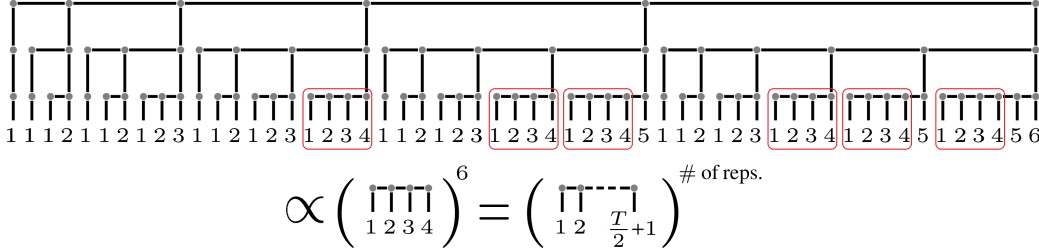

Figure 2: Tensor Network representing the computation of a depth $L = 3$ RAC after $T = 6$ time-steps. See construction in appendix A.

Indeed, we find a specific assignment of the network weights, presented in appendix A.4, for which the Tensor Network effectively takes the form of the basic unit connecting "Start" and "End", raised to the power of the number of its repetitions in the graph (bottom of fig. 2). This basic unit corresponds to a simple computation represented by a grid tensor with Start-End matricization of rank $R$. Raising such a matrix to the Hadamard power of any $p \in \mathbb{Z}$, results in a matrix with a rank upper bounded by $\left(\!\!\binom{R}{p}\!\!\right)$, and the challenge of proving the conjecture amounts to proving that the upper bound is tight in this case. In appendix A.4, we prove that the number of repetitions of the basic unit connecting "Start" and "End" in the deep RAC Tensor Network graph, is exactly equal to $\left(\!\!\binom{T/2}{L-1}\!\!\right)$ for any depth $L$. For example, in the $T = 6$, $L = 3$ network illustrated in fig. 2, the number of repetitions indeed corresponds to $p = \left(\!\!\binom{3}{2}\!\!\right) = 6$. It is noteworthy that for $L = 1, 2$ the bound in conjecture 1 coincides with the bounds that were proved for these depths in theorem 1.

Conjecture 1 indicates that beyond the proved exponential advantage in memory capacity of deep networks over shallow ones, a further exponential separation may be shown between recurrent networks of different depths. We leave the proof of this result, which can reinforce and refine the understanding of advantages brought forth by depth in recurrent networks, as an open problem.

## 5 DISCUSSION

The notion of depth efficiency, by which deep networks efficiently express functions that would require shallow networks to have a super-polynomial size, is well established in the context of convolutional networks. However, recurrent networks differ from convolutional networks, as they are suited by design to tackle inputs of varying lengths. Accordingly, depth efficiency alone does not account for the remarkable performance of recurrent networks on long input sequences. In this paper, we identified a fundamental need for a quantifier of 'time-series expressivity', quantifying the memory capacity of recurrent networks. In order to meet this need, we proposed a measure of the ability of recurrent networks to model long-term temporal dependencies, in the form of the Start-End separation rank. The separation rank was used to quantify correlations in convolutional networks, and has roots in the field of quantum physics. The proposed measure adjusts itself to the temporal extent of the input series, and quantifies the ability of the recurrent network to correlate the incoming sequential data as time progresses.

We analyzed the class of Recurrent Arithmetic Circuits, which are closely related to successful RNN architectures, and proved that the Start-End separation rank of deep RACs increases exponentially as the input sequence extends, while that of shallow RACs is independent of the input length. These results, which demonstrate that depth brings forth an overwhelming advantage in the ability of recurrent networks to model long-term dependencies, were achieved by combining tools from the fields of measure theory, tensorial analysis, combinatorics, graph theory and quantum physics.

Such analyses may be readily extended to other architectural features employed in modern recurrent networks. Indeed, the same time-series expressivity question may now be applied to the different variants of LSTM networks, and the proposed notion of Start-End separation rank may be employed for quantifying their memory capacity. We have demonstrated that such a treatment can go beyond unveiling the origins of the success of a certain architectural choice, and leads to new insights. The above established observation that correlations achievable by vanilla shallow recurrent network do not adapt at all to the sequence length, is an exemplar of this potential.

Moreover, practical recipes may emerge by such theoretical analyses. The experiments preformed in Hermans and Schrauwen (2013), suggest that shallow layers of recurrent networks are related to

short time-scales, *e.g.* in speech: phonemes, syllables, words, while deeper layers appear to support correlations of longer time-scales, *e.g.* full sentences, elaborate questions. These findings open the door to further depth related investigations in recurrent networks, and specifically the role of each layer in modeling temporal correlations may be better understood. Levine et al. (2017) establish theoretical observations which translate into practical conclusions regarding the number of hidden channels to be chosen for each layer in a deep convolutional network. The conjecture presented in this paper, by which the Start-End separation rank of recurrent networks grows exponentially with depth, can similarly entail practical recipes for enhancing their memory capacity. Such analyses can be reinforced by experiments, and lead to a profound understanding of the contribution of deep layers to the recurrent network's memory. Indeed, we view this work as an important step towards novel methods of matching the recurrent network architecture to the temporal correlations in a given sequential data set.

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

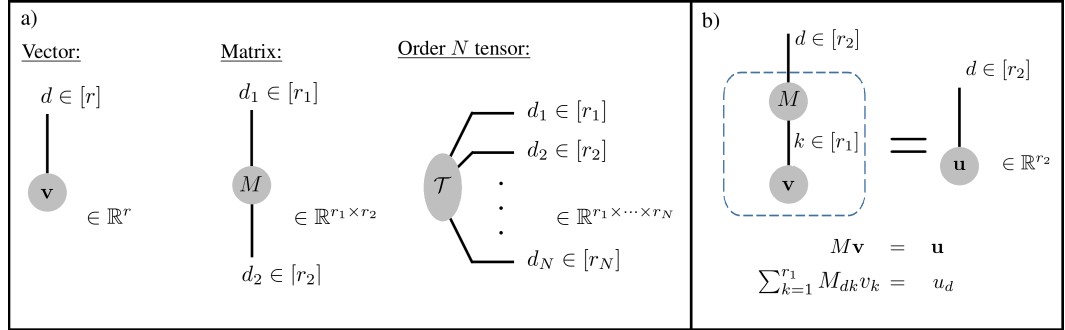

Figure 3: A quick introduction to Tensor Networks (TNs). a) Tensors in the TN are represented by nodes. The degree of the node corresponds to the order of the tensor represented by it. b) A matrix multiplying a vector in TN notation. The contracted index $k$, which connects two nodes, is summed upon, while the open index $d$ is not. The number of open indices equals the order of the tensor represented by the entire network. All of the indices receive values that range between 1 and their bond dimension. The contraction is marked by the dashed line.

## A  TENSOR NETWORK REPRESENTATION OF RECURRENT ARITHMETIC CIRCUITS

In this section, we expand our algebraic view on recurrent networks and make use of a graphical approach to tensor decompositions referred to as Tensor Networks (TNs). The tool of TNs is mainly used in the many-body quantum physics literature for a graphical decomposition of tensors, and has been recently connected to the deep learning field by Levine et al. (2017), who constructed a deep convolutional network in terms of a TN. The use of TNs in machine learning has appeared in an empirical context, where Stoudenmire and Schwab (2016) trained a Matrix Product State (MPS) TN to preform supervised learning tasks on the MNIST data set (LeCun et al., 1998). The constructions presented in this section suggest a separation in expressiveness between recurrent networks of different depths, as formulated by conjecture 1.

We begin in section A.1 by providing a brief introduction to TNs. Next, we present in section A.2 the TN which corresponds to the calculation of a shallow RAC, and tie it to a common TN architecture referred to as a *Matrix Product State* (MPS) (see overview in e.g. Orús (2014)), and equivalently to the *tensor train* (TT) decomposition (Oseledets, 2011). Subsequently, we present in section A.3 a TN construction of a deep RAC, and emphasize the characteristics of this construction that are the origin of the enhanced ability of deep RACs to model elaborate temporal dependencies. Finally, in section A.4, we make use of the above TNs construction in order to formally motivate conjecture 1, according to which the Start-End separation rank of RACs grows exponentially with depth.

### A.1  INTRODUCTION TO TENSOR NETWORKS

A TN is a weighted graph, where each node corresponds to a tensor whose order is equal to the degree of the node in the graph. Accordingly, the edges emanating out of a node, also referred to as its legs, represent the different modes of the corresponding tensor. The weight of each edge in the graph, also referred to as its bond dimension, is equal to the dimension of the appropriate tensor mode. In accordance with the relation between mode, dimension and index of a tensor presented in section 3.2, each edge in a TN is represented by an index that runs between 1 and its bond dimension. Fig. 3(a) shows three examples: (1) A vector, which is a tensor of order 1, is represented by a node with one leg. (2) A matrix, which is a tensor of order 2, is represented by a node with two legs. (3) Accordingly, a tensor of order $N$ is represented in the TN as a node with $N$ legs.

We move on to present the connectivity properties of a TN. Edges which connect two nodes in the TN represent an operation between the two corresponding tensors. A index which represents such an edge is called a contracted index, and the operation of contracting that index is in fact a summation over all of the values it can take. An index representing an edge with one loose end is called an open index. The tensor represented by the entire TN, whose order is equal to the number of open indices, can be calculated by summing over all of the contracted indices in the network. An example for a contraction of a simple TN is depicted in fig. 3(b). There, a TN corresponding to the operation of multiplying a vector $\mathbf{v} \in \mathbb{R}^{r_1}$ by a matrix $M \in \mathbb{R}^{r_2 \times r_1}$ is performed by summing over the only contracted index, $k$. As there is only one open index, $d$, the result of contracting the network is an order 1 tensor (a vector): $\mathbf{u} \in \mathbb{R}^{r_2}$ which upholds $\mathbf{u} = M\mathbf{v}$. Though we use below the contraction of indices in more elaborate TNs, this operation can be essentially viewed as a generalization of matrix multiplication.

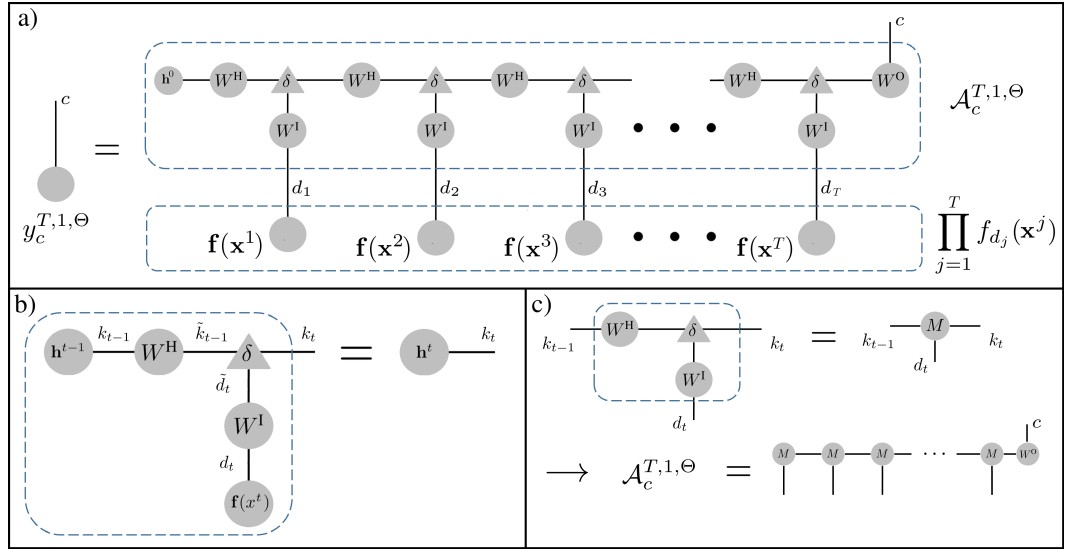

Figure 4: a) The Tensor Network representing the calculation performed by a shallow RAC. b) A Tensor Network construction of the recursive relation given an eq. 1. c) A presentation of the shallow RAC weights tensor in a standard MPS form.

## A.2 SHALLOW RAC TENSOR NETWORK

The computation of the output at time $T$ that is preformed by the shallow recurrent network given by eqs. 1 and 3, or alternatively by eqs. 6 and 7, can be written in terms of a TN. Fig. 4(a) shows this TN, which given some initial hidden state $\mathbf{h}_0$, is essentially a temporal concatenation of a unit cell that preforms a similar computation at every time-step, as depicted in fig. 4(b). For any time $t < T$, this unit cell is composed of the input weights matrix, $W^{\mathrm{I}}$, contracted with the inputs vector, $\mathbf{f}(\mathbf{x}^t)$, and the hidden weights matrix, $W^{\mathrm{H}}$, contracted with the hidden state vector of the previous time-step, $\mathbf{h}^{t-1}$. The final component in each unit cell is the 3 legged triangle representing the order 3 tensor $\delta \in \mathbb{R}^{R \times R \times R}$, referred to as the $\delta$ *tensor*, defined by:

$$\delta_{i_1 i_2 i_3} \equiv \begin{cases} 1, & i_1 = i_2 = i_3 \\ 0, & otherwise \end{cases}, \tag{10}$$

with $i_j \in [R] \ \forall j \in [3]$, *i.e.* its entries are equal to 1 only on the super-diagonal and are zero otherwise. The use of a triangular node in the TN is intended to remind the reader of the restriction given in eq. 10. The recursive relation that is defined by the unit cell, is given by the TN in fig. 4(b):

$$h_{k_t}^t = \sum_{k_{t-1}, \tilde{k}_{t-1}, \tilde{d}_t = 1}^{R} \sum_{d_t = 1}^{M} W_{\tilde{k}_{t-1} k_{t-1}}^{\mathrm{H}} h_{k_{t-1}}^{t-1} W_{\tilde{d}_t d_t}^{\mathrm{I}} f_{d_t}(\mathbf{x}^t) \delta_{\tilde{k}_{t-1} \tilde{d}_t k_t} =$$

$$\sum_{\tilde{k}_{t-1} \tilde{d}_t = 1}^{R} (W^{\mathrm{H}} \mathbf{h}^{t-1})_{\tilde{k}_{t-1}} (W^{\mathrm{I}} \mathbf{f}(\mathbf{x}^t))_{\tilde{d}_t} \delta_{\tilde{k}_{t-1} \tilde{d}_t k_t} = (W^{\mathrm{H}} \mathbf{h}^{t-1})_{k_t} (W^{\mathrm{I}} \mathbf{f}(\mathbf{x}^t))_{k_t}, \tag{11}$$

where $k_t \in [R]$. In the first equality, we simply follow the TN prescription and write a summation over all of the contracted indices in the left hand side of fig. 4(b), in the second equality we use the definition of matrix multiplication, and in the last equality we use the definition of the $\delta$ tensor. The component-wise equality of eq. 11 readily implies $\mathbf{h}^t = (W^{\mathrm{H}} \mathbf{h}^{t-1}) \odot (W^{\mathrm{I}} \mathbf{f}(\mathbf{x}^t))$, reproducing the recursive relation in eqs. 1 and 3, which defines the operation of the shallow RAC. From the above treatment, it is evident that the restricted $\delta$ tensor is in fact the component in the TN that yields the element-wise multiplication property. After $T$ repetitions of the unit cell calculation with the sequential input $\{\mathbf{x}^t\}_{t=1}^T$, a final multiplication of the hidden state vector $\mathbf{h}^T$ by the output weights matrix $W^{\mathrm{O}}$ yields the output vector $\mathbf{y}^{T,1,\Theta}$.

The tensor network which represents the order $T$ shallow RAC weights tensor $\mathcal{A}_c^{T,1,\Theta}$, which appears in eqs. 6 and 7, is given by the TN in the upper part of fig. 4(a). In fig. 4(c), we show that by a simple contraction of indices, the TN representing the shallow RAC weights tensor $\mathcal{A}_c^{T,1,\Theta}$ can be drawn in the form of a standard MPS TN. This TN allows the representation of an order $T$ tensor with a linear (in $T$) amount of parameters, rather than the regular exponential amount ($\mathcal{A}$ has $M^T$ entries). The decomposition which corresponds to this TN is known as the Tensor Train (TT) decomposition of rank $R$ in the tensor analysis community, its explicit form given in eq. 7.

The presentation of the shallow recurrent network in terms of a TN allows the employment of the min-cut analysis, which was introduced by Levine et al. (2017) in the context of convolutional networks, for quantification of the information flow across time modeled by the shallow recurrent network. This was indeed preformed in our proof of the shallow case of theorem 1. We now move on to present the computation preformed by a deep recurrent network in the language of TNs.

### A.3 DEEP RAC TENSOR NETWORK

The construction of a TN which matches the calculation of a deep recurrent network is far less trivial than that of the shallow case, due to the seemingly innocent property of reusing information which lies at the heart of the calculation of deep recurrent networks. Specifically, all of the hidden states of the network are reused, since the state of each layer at every time-step is duplicated and sent as an input to the calculation of the same layer in the next time-step, and also as an input to the next layer up in the same time-step (see fig. 1(b)). The required operation of duplicating a vector and sending it to be part of two different calculations, which is simply achieved in any practical setting, is actually impossible to represent in the framework of TNs. We formulate this notion in the following claim:

**Claim 3.** *Let $v \in \mathbb{R}^P, P \in \mathbb{N}$ be a vector. $v$ is represented by a node with one leg in the TN notation. The operation of duplicating this node,* i.e. *forming two separate nodes of degree 1, each equal to $v$, cannot be achieved by any TN.*

*Proof.* We assume by contradiction that there exists a Tensor Network $\phi$ which operates on any vector $v \in \mathbb{R}^P$ and clones it to two separate nodes of degree 1, each equal to $v$, to form an overall TN representing $v \otimes v$. Component wise, this implies that $\phi$ upholds $\forall v \in \mathbb{R}^P : \sum_{i=1}^{P} \phi_{ijk} v_i = v_j v_k$. By our assumption, $\phi$ duplicates the standard basis elements of $\mathbb{R}^P$, denoted $\{\hat{\mathbf{e}}^{(\alpha)}\}_{\alpha=1}^{P}$, meaning that $\forall \alpha \in [P]$:

$$\sum_{i=1}^{P} \phi_{ijk} \hat{e}_i^{(\alpha)} = \hat{e}_j^{(\alpha)} \hat{e}_k^{(\alpha)}. \tag{12}$$

By definition of the standard basis elements, the left hand side of eq. 12 takes the form $\phi_{\alpha jk}$ while the right hand side equals 1 only if $j = k = \alpha$, and otherwise 0. Utilizing the $\delta$-tensor notation presented in eq. 10, in order to successfully clone the standard basis elements, eq. 12 implies that $\phi$ must uphold $\phi_{\alpha jk} = \delta_{\alpha jk}$. However, for $v = \mathbf{1}$, i.e. $\forall j \in [P] : v_j = 1$, a cloning operation does not take place when using this value of $\phi$, since $\sum_{i=1}^{P} \phi_{ijk} v_i = \sum_{i=1}^{P} \delta_{ijk} = \delta_{jk} \neq 1 = v_i v_j$, in contradiction to $\phi$ duplicating any vector in $\mathbb{R}^P$. $\square$

Claim 3 seems to pose a hurdle in our pursuit of a TN representing a deep recurrent network. Nonetheless, a form of such a TN may be attained by a simple 'trick' – in order to model the duplication that is inherently present in the deep recurrent network computation, we resort to duplicating the input data itself. By this technique, for every duplication that takes place along the calculation, the input is inserted into the TN multiple times, once for each sequence that leads to the duplication point. This principle, which allows us to circumvent the restriction imposed by claim 3, yields the elaborate TN construction of deep RACs depicted in fig. 5.

It is important to note that these TNs, which grow exponentially in size as the depth $L$ of the recurrent network represented by them increases, are merely a theoretical tool for analysis and not a suggested implementation scheme for deep recurrent networks. The actual deep recurrent network is constructed according to the simple scheme given in fig. 1(b), which grows only linearly in size as the depth $L$ increases, despite the corresponding TN growing exponentially. In fact, this exponential 'blow-up' in the size of the TNs representing the deep recurrent networks is closely related to their ability to model more intricate correlations over longer periods of time in comparison with their shallower counterparts, which was established in section 4.

Fig. 5 shows TNs which correspond to depth $L = 2, 3$ RACs. Even though the TNs in fig. 5 seem rather convoluted and complex, their architecture follows clear recursive rules. In fig. 5(a), a depth $L = 2$ recurrent network is presented, spread out in time onto $T = 4$ time-steps. To understand the logic underlying the input duplication process, which in turn entails duplication of entire segments of the TN, we focus on the calculation of the hidden state vector $\mathbf{h}^{2,2}$ that is presented in fig. 5(b). When the first inputs vector, $\mathbf{f}(\mathbf{x}^1)$, is inserted into the network, it is multiplied by $W^{I,1}$ and the outcome is equal to $\mathbf{h}^{1,1}$.[4] Next, $\mathbf{h}^{1,1}$ is used in two different places, as an inputs vector to layer $L = 2$ at time $t = 1$, and as a hidden state vector in layer $L = 1$ for time $t = 2$ calculation. Our input duplication technique inserts $\mathbf{f}(\mathbf{x}^1)$ into the network twice, so that the same exact $\mathbf{h}^{1,1}$ is achieved twice in the TN, as marked by the red dotted line in fig. 5(b). This way, every copy of $\mathbf{h}^{1,1}$

---

[4]In this figure, the initial condition for each layer $l \in L$, $\mathbf{h}^{l,0}$, is chosen such that a vector of ones will be present in the initial element-wise multiplication: $(\mathbf{h}^{0,l})^T = \mathbf{1}^T (W^{H,l})^{\dagger}$, where $\dagger$ denotes the pseudoinverse operation.

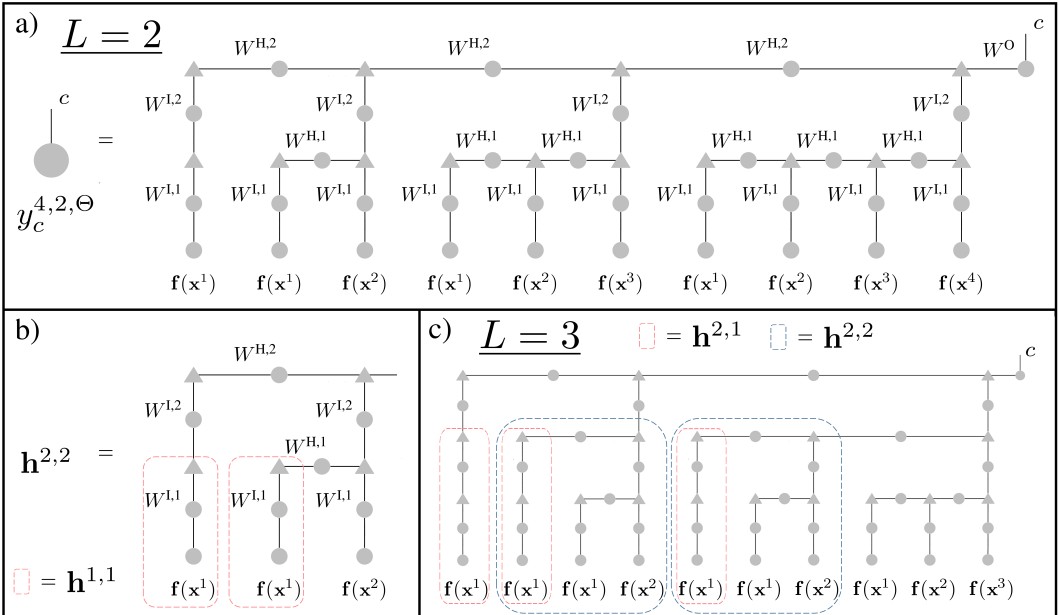

Figure 5: a) The Tensor Network representing the calculation preformed by a depth $L = 2$ RAC after 4 time-steps. b) A Tensor Network construction of the hidden state $\mathbf{h}^{2,2}$ (see eq. 13), which involves duplication of the hidden state $\mathbf{h}^{1,1}$ that is achieved by duplicating the input $x^1$. c) The Tensor Network representing the calculation preformed by a depth $L = 3$ RAC after 3 time-steps. Here too, as in any deep RAC, several duplications take place.

goes to the appropriate segment of the calculation, and indeed the TN in fig. 5(b) holds the correct value of $\mathbf{h}^{2,2}$:

$$\mathbf{h}^{2,2} = \left(W^{\mathrm{H},2}W^{\mathrm{I},2}\mathbf{h}^{1,1}\right) \odot \left(W^{\mathrm{I},2}((W^{\mathrm{H},1}\mathbf{h}^{1,1}) \odot (W^{\mathrm{I},1}\mathbf{f}(\mathbf{x}^2)))\right). \tag{13}$$

The extension to deeper layers leads us to a fractal structure of the TNs, involving many self similarities, as in the $L = 3$ example given in fig. 5(c). The duplication of intermediate hidden states, marked in red and blue in this example, is the source of the apparent complexity of this $L = 3$ RAC TN. Generalizing the above $L = 1, 2, 3$ examples, a TN representing an RAC of general depth $L$ and of $T$ time-steps, would involve in its structure $T$ duplications of TNs representing RACs of depth $L - 1$, each of which has a distinct length in time-steps $i$, where $i \in [T]$. This fractal structure leads to an increasing with depth complexity of the TN representing the depth $L$ RAC computation, which we show in the next subsection to motivate the combinatorial lower bound on the Start-End separation rank of deep RACs, given in conjecture 1.

### A.4 A FORMAL MOTIVATION FOR CONJECTURE 1

The above presented construction of TNs which correspond to deep RACs, allows us to further investigate the effect of network depth on its ability to model long-term temporal dependencies. We present below a formal motivation for the lower bound on the Start-End separation rank of deep recurrent networks, given in conjecture 1. Though our analysis employs TNs visualizations, it is formal nonetheless – these graphs represent the computation in a well-defined manner (see sections A.1-A.3).

Our conjecture relies on the fact that it is sufficient to find a specific instance of the network parameters $\Theta \times \mathbf{h}^{0,l}$ for which $[\![\mathcal{A}(y_c^{T,L,\Theta})]\!]_{S,E}$ achieves a certain rank, in order for this rank to be a lower bound on the Start-End separation rank of the network. This follows from combining claim 2 and lemma 1. Claim 2 assures us that the Start-End separation rank of the function realized by an RAC of any depth $L$, is lower bounded by the rank of the matrix obtained by the corresponding grid tensor matricization: $\mathrm{sep}_{(S,E)}\left(y_c^{T,L,\Theta}\right) \geq$ rank $\left([\![\mathcal{A}(y_c^{T,L,\Theta})]\!]_{S,E}\right)$. Thus, one must show that rank $\left([\![\mathcal{A}(y_c^{T,L,\Theta})]\!]_{S,E}\right) \geq \left(\binom{\min\{M,R\}}{\binom{T/2}{L-1}}\right)$ for all of the values of parameters $\Theta \times \mathbf{h}^{0,l}$ but a set of Lebesgue measure zero, in order to establish the lower bound in conjecture 1. Next, we rely on lemma 1, which states that since the entries of $\mathcal{A}(y_c^{T,L,\Theta})$ are polynomials in the deep recurrent network's weights, it suffices to find a single example for which the rank of the matricized grid

Figure 6: Above: TN representing the computation of a depth $L = 3$ RAC after $T = 6$ time-steps, when choosing $W^{I,2}$ to be of rank-1. See full TN, for general values of the weight matrices, in fig. 2. Below: Reduction of this TN to the factors affecting the Start-End matricization of the grid tensor represented by the TN.

tensor is greater than the desired lower bound. Finding such an example would indeed imply that for almost all of the values of the network parameters, the desired inequality holds.

In the following, we choose a weight assignment that effectively 'separates' between the first layer and higher layers, in the sense that $W^{I,2}$ is of rank-1. This is done in similar spirit to the assignment used in the proof of theorem 1, in which $W^{I,2}_{ij} \equiv \delta_{i1}$ (see section B.3). Under this simplifying assignment, which suffices for our purposes according to the above discussion, the entire computation performed in deeper layers contributes only a constant factor to the matricized grid tensor. In this case, the example of the TN corresponding to an RAC of depth $L = 3$ after $T = 6$ time-steps, which is shown in full in fig.2, takes the form shown in the upper half of fig. 6. Next, in order to evaluate $\text{rank}\left(\llbracket \mathcal{A}(y_c^{T,L,\Theta}) \rrbracket_{S,E}\right)$, we note that graph segments which involve only indices from the "Start" set, will not affect the rank of the matrix under mild conditions on $W^{I,1}, W^{H,1}$.[5] Specifically, under the Start-End matricization these segments will amount to a different constant multiplying each row of the matrix. For the example of the RAC of depth $L = 3$ after $T = 6$ time-steps, this amounts to the effective TN given in the bottom left side of fig. 6. Finally, the dependence of this TN on the indices of time-steps $\{T/2+2, \ldots, T\}$, namely those outside of the basic unit involving indices of time-steps $\{1, \ldots, T/2+1\}$, may only increase the resulting Start-End matricization rank.[6] Thus, we are left with an effective TN resembling the one shown in section 4.2, where the basic unit separating "Start" and "End" indices is raised to the power of the number of its repetitions in the graph. In the following, we prove a claim according to which the number of repetitions of this basic unit in the TN graph increases exponentially with the depth of the RAC:

**Claim 4.** *Let $\phi(T, L, R)$ be the TN representing the computation performed after $T$ time-steps by an RAC with $L$ layers and $R$ hidden channels per layer. Then, the number of occurrences in layer $L = 1$ of the basic unit connecting "Start" and "End" indices (bottom right in fig. 6), is exactly $\left(\!\!\binom{T/2}{L-1}\!\!\right)$.*

*Proof.* Let $y_c^{T,L,\Theta}$ be the function computing the output after $T$ time-steps of an RAC with $L$ layers, $R$ hidden channels per layer and weights denoted by $\Theta$. In order to focus on repetitions in layer $L = 1$, we assign $W^{I,2}_{ij} \equiv \delta_{i1}$ for which the following upholds[7]:

$$\mathcal{A}(y_c^{T,L,\Theta})_{d_1,\ldots,d_T} = (Const.) \prod_{t_L=1}^{T} \prod_{t_{L-1}=1}^{t_L} \cdots \prod_{t_2=1}^{t_3} \sum_{r_1,\ldots,r_{t_2}=1}^{R} \left( \prod_{j=1}^{t_2} W^{I,1}_{r_j d_j} \prod_{j=1}^{t_2-1} W^{H,1}_{r_j r_{j+1}} \right)$$

$$= (Const.)\,(V_{d_1 \ldots d_{T/2}}) \prod_{t_L=T/2+1}^{T} \prod_{t_{L-1}=T/2+1}^{t_L} \cdots \prod_{t_2=T/2+1}^{t_3} \sum_{r_1,\ldots,r_{t_2}=1}^{R} \left( \prod_{j=1}^{t_2} W^{I,1}_{r_j d_j} \prod_{j=1}^{t_2-1} W^{H,1}_{r_j r_{j+1}} \right),$$

where the constant term in the first line is the contribution of the deeper layers under this assignment, and the tensor $V_{d_1 \ldots d_{T/2}}$, which becomes a vector under the Start-End matricization, reflects the contribution of the "Start" set indices. Observing the argument of the chain of products in the above expression, $\sum_{r_1,\ldots,r_{t_2}=1}^{R} \left( \prod_{j=1}^{t_2} W^{I,1}_{r_j d_j} \prod_{j=1}^{t_2-1} W^{H,1}_{r_j r_{j+1}} \right)$, it is an order $t_2$ tensor, exactly given by the TN representing the computation of a depth $L = 1$ RAC after $t_2$ time-steps. Specifically, for $t_2 = T/2 + 1$, it is exactly equal to the basic TN unit connecting "Start" and "End" indices, and for $T/2 + 1 < t_2 \leq T$ it contains this basic unit. This means that in order to obtain the number of repetition of this basic unit in $\phi$, we must count the number of

---

[5]For example, this holds if $W^{I,1}$ is fully ranked and does not have vanishing elements, and $W^{H,1} = I$.

[6]This is not true for any TN of this shape but holds due to the temporal invariance of the recurrent network's weights.

[7]See a similar and more detailed derivation in section B.3.

multiplications implemented by the chain of products in the above expression. Indeed this number is equal to:

$$\sum_{t_L=T/2+1}^{T} \sum_{t_{L-1}=T/2+1}^{t_L} \cdots \sum_{t_2=T/2+1}^{t_3} t_2 = \left(\!\!\left(\begin{array}{c} T/2 \\ L-1 \end{array}\right)\!\!\right)$$

$\square$

Finally, the form of the lower bound presented in conjecture 1 is obtained by considering a rank $R$ matrix, such as the one obtained by the Start-End matricization of the TN basic unit discussed above, raised to the Hadamard power of $\left(\!\!\left(\begin{array}{c} T/2 \\ L-1 \end{array}\right)\!\!\right)$. The rank of the resultant matrix, is upper bounded by $\left(\!\!\left(\begin{array}{c} R \\ \left(\!\!\left(\begin{array}{c} T/2 \\ L-1 \end{array}\right)\!\!\right) \end{array}\right)\!\!\right)$. We leave it as an open problem to prove conjecture 1, by proving that the upper bound is indeed tight in this case.

## B DEFERRED PROOFS

### B.1 PROOF OF CLAIM 1

We begin by proving the equality $\text{sep}_{(S,E)}\left(y_c^{T,1,\Theta}\right) = \text{rank}\left(\llbracket \mathcal{A}_c^{T,1,\Theta} \rrbracket_{S,E}\right)$. As shown in Cohen and Shashua (2017), for any function $f : R^s \times \cdots \times R^s \to R$ which follows the structure of eq. 6 with a general weights tensor $\mathcal{A}$, assuming that $\{f_d\}_{d=1}^{M}$ are linearly independent, measurable, and square-integrable (as assumed in claim 1), it holds that $\text{sep}_{(S,E)}(f) = \text{rank}(\llbracket \mathcal{A} \rrbracket_{S,E})$. Specifically, for $f = y_c^{T,1,\Theta}$ and $\mathcal{A} = \mathcal{A}_c^{T,1,\Theta}$ the above equality holds.

It remains to prove that there exists template vectors for which $\text{rank}\left(\llbracket \mathcal{A}_c^{T,1,\Theta} \rrbracket_{S,E}\right) = \text{rank}\left(\llbracket \mathcal{A}(y_c^{T,1,\Theta}) \rrbracket_{S,E}\right)$. For any given set of template vectors $\mathbf{x}^{(1)}, \ldots, \mathbf{x}^{(M)} \in \mathcal{X}$, we define the matrix $F \in \mathbb{R}^{M \times M}$ such that $F_{ij} = f_j(\mathbf{x}^{(i)})$, for which it holds that:

$$\mathcal{A}(y_c^{T,1,\Theta})_{k_1,\ldots,k_T} = \sum_{d_1,\ldots,d_T=1}^{M} \left(\mathcal{A}_c^{T,1,\Theta}\right)_{d_1,\ldots,d_T} \prod_{i=1}^{T} f_{d_i}(\mathbf{x}^{(k_i)})$$

$$= \sum_{d_1,\ldots,d_T=1}^{M} \left(\mathcal{A}_c^{T,1,\Theta}\right)_{d_1,\ldots,d_T} \prod_{i=1}^{T} F_{k_i d_i}.$$

The right-hand side in the above equation can be regarded as a linear transformation of $\mathcal{A}_c^{T,1,\Theta}$ specified by the tensor operator $F \otimes \cdots \otimes F$, which is more commonly denoted by $(F \otimes \cdots \otimes F)(\mathcal{A}_c^{T,1,\Theta})$. According to lemma 5.6 in Hackbusch (2012), if $F$ is non-singular then $\text{rank}\left(\llbracket (F \otimes \cdots \otimes F)(\mathcal{A}_c^{T,1,\Theta}) \rrbracket_{I,J}\right) = \text{rank}\left(\llbracket \mathcal{A}_c^{T,1,\Theta} \rrbracket_{I,J}\right)$, for any partition $(I,J)$. To conclude our proof, we simply note that Cohen and Shashua (2016) showed that if $\{f_d\}_{d=1}^{M}$ are linearly independent then there exists template vectors for which $F$ is non-singular.

$\square$

### B.2 PROOF OF CLAIM 2

If $\text{sep}_{(S,E)}\left(y_c^{T,L,\Theta}\right) = \infty$ then the inequality is trivially satisfied. Otherwise, assume that $\text{sep}_{(S,E)}\left(y_c^{T,L,\Theta}\right) = K \in \mathbb{N}$, and let $\{g_i^s, g_i^e\}_{i=1}^{K}$ be the functions of the respective decomposition to a sum of separable functions, i.e. that the following holds:

$$y_c^{T,L,\Theta}(\mathbf{x}^1,\ldots,\mathbf{x}^T) = \sum_{\nu=1}^{K} g_\nu^s(\mathbf{x}^1,\ldots,\mathbf{x}^{T/2}) \cdot g_\nu^e(\mathbf{x}^{T/2+1},\ldots,\mathbf{x}^T).$$

Then, by definition of the grid tensor, for any template vectors $\mathbf{x}^{(1)}, \ldots, \mathbf{x}^{(M)} \in \mathcal{X}$ the following equality holds:

$$\mathcal{A}(y_c^{T,L,\Theta})_{d_1,\ldots,d_N} = \sum_{\nu=1}^{K} g_\nu^s(\mathbf{x}^{(d_1)},\ldots,\mathbf{x}^{(d_{T/2})}) \cdot g_\nu^e(\mathbf{x}^{(d_{T/2+1})},\ldots,\mathbf{x}^{(d_T)})$$

$$\equiv \sum_{\nu=1}^{K} V_{d_1,\ldots,d_{T/2}}^{\nu} U_{d_{T/2+1},\ldots,d_T}^{\nu},$$

where $V^\nu$ and $U^\nu$ are the tensors holding the values of $g_\nu^s$ and $g_\nu^e$, respectively, at the points defined by the template vectors. Under the matricization according to the $(S, E)$ partition, it holds that $[\![V^\nu]\!]_{S,E}$ and $[\![U^\nu]\!]_{S,E}$ are column and row vectors, respectively, which we denote by $\mathbf{v}_\nu$ and $\mathbf{u}_\nu^T$. It follows that the matricization of the grid tensor is given by:

$$[\![\mathcal{A}(y_c^{T,L,\Theta})]\!]_{S,E} = \sum_{\nu=1}^{K} \mathbf{v}_\nu \mathbf{u}_\nu^T,$$

which means that $\mathrm{rank}\left([\![\mathcal{A}(y_c^{T,L,\Theta})]\!]_{S,E}\right) \leq K = \mathrm{sep}_{(S,E)}\left(y_c^{T,L,\Theta}\right)$.

$\square$

### B.3 PROOF OF THEOREM 1

In this sub-section, we follow the proof strategy that is outlined in section 4, and prove theorem 1, which shows an exponential advantage of deep recurrent networks over shallow ones in the ability to model long-term dependencies, as measured by the Start-End separation rank (see section 3.1). In sections B.3.1 and B.3.2, we prove the bounds on the Start-End separation rank of the shallow and deep RACs, respectively, while more technical lemmas which are employed during the proof are relegated to section B.3.3.

### B.3.1 THE START-END SEPARATION RANK OF SHALLOW RACS

We consider the Tensor Network construction of the calculation carried out by a shallow RAC, given in fig. 4. According to the presented construction, the shallow RAC weights tensor (eqs. 6 and 7) is represented by a Matrix Product State (MPS) Tensor Network (Orús, 2014), with the following order-3 tensor building block: $M_{k_{t-1}d_t k_t} = W^{\mathrm{I}}_{k_t d_t} W^{\mathrm{H}}_{k_t k_{t-1}}$, where $d_t \in [M]$ is the input index and $k_{t-1}, k_t \in [R]$ are the internal indices (see fig. 4(c)). In TN terms, this means that the bond dimension of this MPS is equal to $R$. We apply the result of Levine et al. (2017), who state that the rank of the matrix obtained by matricizing any tensor according to a partition $(S, E)$ is equal to a min-cut separating $S$ from $E$ in the Tensor Network graph representing this tensor, for all of the values of the TN parameters but a set of Lebesgue measure zero. In this MPS Tensor Network, the minimal cut w.r.t. the partition $(S, E)$ is equal to the bond dimension $R$, unless $R > M^{T/2}$, in which case the minimal cut contains the external legs instead. Thus, in the TN representing $\mathcal{A}^{T,1,\Theta}_c$, the minimal cut w.r.t. the partition $(S, E)$ is equal to $\min\{R, M^{T/2}\}$, implying $\mathrm{rank}\left(\llbracket\mathcal{A}^{T,1,\Theta}_c\rrbracket\right)_{S,E} = \min\{R, M^{T/2}\}$ for all values of the parameters but a set of Lebesgue measure zero. The first half of the theorem follows from applying claim 1, which assures us that the Start-End separation rank of the function realized by a shallow ($L = 1$) RAC is equal to $\mathrm{rank}\left(\llbracket\mathcal{A}^{T,1,\Theta}_c\rrbracket\right)_{S,E}$.

$\square$

### B.3.2 LOWER-BOUND ON THE START-END SEPARATION RANK OF DEEP RACS

For a deep network, claim 2 assures us that the Start-End separation rank of the function realized by a depth $L = 2$ RAC is lower bounded by the rank of the matrix obtained by the corresponding grid tensor matricization, for any choice of template vectors. Specifically:

$$\mathrm{sep}_{(S,E)}\left(y^{T,L,\Theta}_c\right) \geq \mathrm{rank}\left(\llbracket\mathcal{A}(y^{T,L,\Theta}_c)\rrbracket_{S,E}\right).$$

Thus, since it trivially holds that $\mathrm{rank}\left(\llbracket\mathcal{A}(y^{T,L,\Theta}_c)\rrbracket_{S,E}\right) \leq M^{T/2}$ (the rank is smaller than the dimension of the matrix), proving that $\mathrm{rank}\left(\llbracket\mathcal{A}(y^{T,L,\Theta}_c)\rrbracket_{S,E}\right) \geq \left(\binom{\min\{R,M\}}{T/2}\right)$ for all of the values of parameters $\Theta \times \mathbf{h}^{0,l}$ but a set of Lebesgue measure zero, would satisfy the theorem.

In the following, we provide an assignment of weight matrices and initial hidden states for which $\mathrm{rank}\left(\llbracket\mathcal{A}(y^{T,L,\Theta}_c)\rrbracket_{S,E}\right) = \left(\binom{\min\{R,M\}}{T/2}\right)$. In accordance with claim 5, this will suffice as such an assignment implies this rank is achieved for all configurations of the recurrent network weights but a set of Lebesgue measure zero.

We begin by choosing a specific set of template vectors $\mathbf{x}^{(1)}, \ldots, \mathbf{x}^{(M)} \in \mathcal{X}$. Let $F \in \mathbb{R}^{M \times M}$ be a matrix with entries defined by $F_{ij} \equiv f_j(\mathbf{x}^{(i)})$. According to Cohen and Shashua (2016), since $\{f_d\}_{d=1}^M$ are linearly independent, then there is a choice of template vectors for which $F$ is non-singular.

Next, we describe our assignment. In the expressions below we use the notation $\delta_{ij} = \begin{cases} 1 & i = j \\ 0 & i \neq j \end{cases}$. Let $z \in \mathbb{R} \setminus \{0\}$ be an arbitrary non-zero real number, let $\Omega \in \mathbb{R}_+$ be an arbitrary positive real number, and let $Z \in \mathbb{R}^{R \times M}$ be a matrix with entries $Z_{ij} \equiv \begin{cases} z^{\Omega^i \delta_{ij}} & i \leq M \\ 0 & i > M \end{cases}$.

We set $W^{\mathrm{I},1} \equiv Z \cdot (F^T)^{-1}$ and set $W^{\mathrm{I},2}$ such that its entries are $W^{\mathrm{I},2}_{ij} \equiv \delta_{i1}$, we set $W^{\mathrm{H},1} \equiv W^{\mathrm{H},2} \equiv I$, i.e. to the identity matrix, and additionally we set the entries of $W^{\mathrm{O}}$ to $W^{\mathrm{O}}_{ij} = \delta_{1j}$. Finally, we choose the initial hidden state values so they bear no effect on the calculation, namely $\mathbf{h}^{0,l} = \left(W^{\mathrm{H},l}\right)^{-1} \mathbf{1} = \mathbf{1}$ for $l = 1, 2$.

Under the above assignment, the output for the corresponding class $c$ after $T$ time-steps is equal to:

$$y_c^{T,L,\Theta}(\mathbf{x}^1,\ldots,\mathbf{x}^T) = \left(W^{\mathrm{O}}\mathbf{h}^{T,2}\right)_c$$
$$(W_{ij}^{\mathrm{O}} \equiv \delta_{1j}) \Rightarrow = (\mathbf{h}^{T,2})_1$$
$$(\text{eq. 4}) \Rightarrow = \left((W^{\mathrm{H},2}\mathbf{h}^{T-1,2}) \odot (W^{\mathrm{I},2}\mathbf{h}^{T,1})\right)_1$$
$$(W^{\mathrm{H},2} \equiv I) \Rightarrow = \left((\mathbf{h}^{T-1,2}) \odot (W^{\mathrm{I},2}\mathbf{h}^{T,1})\right)_1$$
$$(\mathbf{h}^{0,2} = \mathbf{1}) \Rightarrow = \prod_{t=1}^{T}\left(W^{\mathrm{I},2}\mathbf{h}^{t,1}\right)_1$$
$$(W_{ij}^{\mathrm{I},2} \equiv \delta_{1i}) \Rightarrow = \prod_{t=1}^{T}\sum_{r=1}^{R}(\mathbf{h}^{t,1})_r$$
$$(\text{eq. 4}) \Rightarrow = \prod_{t=1}^{T}\sum_{r=1}^{R}\left((W^{\mathrm{H},1}\mathbf{h}^{t-1,1}) \odot (W^{\mathrm{I},1}\mathbf{f}(\mathbf{x}^t))\right)_r$$
$$(W^{\mathrm{H},1} \equiv I) \Rightarrow = \prod_{t=1}^{T}\sum_{r=1}^{R}\left((\mathbf{h}^{t-1,1}) \odot (W^{\mathrm{I},1}\mathbf{f}(\mathbf{x}^t))\right)_r$$
$$(\mathbf{h}^{0,1} = \mathbf{1}) \Rightarrow = \prod_{t=1}^{T}\sum_{r=1}^{R}\prod_{j=1}^{t}\left(W^{\mathrm{I},1}\mathbf{f}(\mathbf{x}^j)\right)_r.$$

When evaluating the grid tensor for our chosen set of template vectors, i.e. $\mathcal{A}(y_c^{T,L,\Theta})_{d_1,\ldots,d_T} = y_c^{T,L,\Theta}(\mathbf{x}^{(d_1)},\ldots,\mathbf{x}^{(d_T)})$, we can substitute $f_j(\mathbf{x}^{(i)}) \equiv F_{ij}$, and thus

$$(W^{\mathrm{I},1}\mathbf{f}(\mathbf{x}^{(d)}))_r = (W^{\mathrm{I},1}F^T)_{rd} = (Z \cdot (F^T)^{-1}F^T)_{rd} = Z_{rd}.$$

Since we defined $Z$ such that for $r \geq \min\{R,M\}$ $Z_{rd} = 0$, and denoting $\bar{R} \equiv \min\{R,M\}$ for brevity of notation, the grid tensor takes the following form:

$$\mathcal{A}(y_c^{T,L,\Theta})_{d_1,\ldots,d_T} = \prod_{t=1}^{T}\sum_{r=1}^{\bar{R}}\prod_{j=1}^{t}Z_{rd_j} = \left(\prod_{t=1}^{T/2}\sum_{r=1}^{\bar{R}}\prod_{j=1}^{t}Z_{rd_j}\right) \cdot \left(\prod_{t=T/2+1}^{T}\sum_{r=1}^{\bar{R}}\prod_{j=1}^{t}Z_{rd_j}\right),$$

where we split the product into two expressions, the left part that contains only the indices in the start set $S$, i.e. $d_1,\ldots,d_{T/2}$, and the right part which contains all external indices (in the start set $S$ and the end set $E$). Thus, under matricization w.r.t. the Start-End partition, the left part is mapped to a vector $\mathbf{a} \equiv \left[\!\left[\prod_{t=1}^{T/2}\sum_{r=1}^{\bar{R}}\prod_{j=1}^{t}Z_{rd_j}\right]\!\right]_{S,E}$ containing only non-zero entries per the definition of $Z$, and the right part is mapped to a matrix $B \equiv \left[\!\left[\prod_{t=T/2+1}^{T}\sum_{r=1}^{\bar{R}}\prod_{j=1}^{t}Z_{rd_j}\right]\!\right]_{S,E}$, where each entry of $\mathbf{u}$ multiplies the corresponding row of $B$. This results in:

$$\left[\!\left[\mathcal{A}(y_c^{T,L,\Theta})_{d_1,\ldots,d_T}\right]\!\right]_{S,E} = \mathrm{diag}(\mathbf{a}) \cdot B.$$

Since $\mathbf{a}$ contains only non-zero entries, $\mathrm{diag}(\mathbf{a})$ is of full rank, and so $\mathrm{rank}\left(\left[\!\left[\mathcal{A}(y_c^{T,L,\Theta})_{d_1,\ldots,d_T}\right]\!\right]_{S,E}\right) = \mathrm{rank}(B)$, leaving us to prove that $\mathrm{rank}(B) = \left(\!\!\left(\begin{smallmatrix}\bar{R}\\T/2\end{smallmatrix}\right)\!\!\right)$. For brevity of notation, we define $N \equiv \left(\!\!\left(\begin{smallmatrix}\bar{R}\\T/2\end{smallmatrix}\right)\!\!\right)$

To prove the above, it is sufficient to show that $B$ can be written as a sum of $N$ rank-1 matrices, i.e. $B = \sum_{i=1}^{N}\mathbf{u}^{(i)} \otimes \mathbf{v}^{(i)}$, and that $\{\mathbf{u}^{(i)}\}_{i=1}^{N}$ and $\{\mathbf{v}^{(i)}\}_{i=1}^{N}$ are two sets of linearly independent vectors. Indeed, applying claim 6 on the entries of $B$, specified w.r.t. the row $(d_1,\ldots,d_{T/2})$ and column $(d_{T/2+1},\ldots,d_T)$, yields the following form:

$$B_{(S,E)} = \sum_{\mathbf{p}^{(T/2)}\in\text{states}\left(\bar{R},T/2\right)}\left(\prod_{r=1}^{\bar{R}}\prod_{j=1}^{T/2}Z_{rd_j}^{p_r^{(T/2)}}\right) \cdot \left(\sum_{\substack{(\mathbf{p}^{(T/2-1)},\ldots,\mathbf{p}^{(1)})\\\in\text{trajectory}\left(\mathbf{p}^{(T/2)}\right)}}\prod_{r=1}^{\bar{R}}\prod_{j=T/2+1}^{T}Z_{rd_j}^{p_r^{(T-j+1)}}\right),$$

where for all $k$, $\mathbf{p}^{(k)}$ is $\bar{R}$-dimensional vector of non-negative integer numbers which sum to $k$, and we explicitly define states $\left(\bar{R},T/2\right)$ and trajectory $\left(\mathbf{p}^{(T/2)}\right)$ in claim 6, providing a softer more intuitive definition

hereinafter. $\text{states}\left(\bar{R}, {}^T\!/_2\right)$ can be viewed as the set of all possible states of a bucket containing ${}^T\!/_2$ balls of $\bar{R}$ colors, where $p_r^{(T/2)}$ for $r \in [\bar{R}]$ specifies the number of balls of the $r$'th color. $\text{trajectory}\left(\mathbf{p}^{(T/2)}\right)$ can be viewed as all possible trajectories from a given state to an empty bucket, i.e. $(0, \ldots, 0)$, where at each step we remove a single ball from the bucket. We note that the number of all initial states of the bucket is exactly $\left|\text{states}\left(\bar{R}, {}^T\!/_2\right)\right| = N \equiv \left(\!\!\binom{\bar{R}}{{}^T\!/_2}\!\!\right)$. Moreover, since the expression in the left parentheses contains solely indices from the start set $S$, i.e. $d_1, \ldots, d_{T/2}$, while the right contains solely indices from the end set $E$, i.e. $d_{T/2+1}, \ldots, d_T$, then each summand is in fact a rank-1 matrix. Specifically, it can be written as $\mathbf{u}^{\mathbf{p}^{(T/2)}} \otimes \mathbf{v}^{\mathbf{p}^{(T/2)}}$, where the entries of $\mathbf{u}^{\mathbf{p}^{(T/2)}}$ are represented by the expression in the left parentheses, and those of $\mathbf{v}^{\mathbf{p}^{(T/2)}}$ by the expression in the right parentheses.

We prove that the set $\left\{\mathbf{u}^{\mathbf{p}^{(T/2)}} \in \mathbb{R}^{M^{T/2}}\right\}_{\mathbf{p}^{(T/2)} \in \text{states}\left(\bar{R}, T/2\right)}$ is linearly independent by arranging it as the columns of the matrix $U \in \mathbb{R}^{M^{T/2} \times N}$, and showing that its rank equals to $N$. Specifically, we observe the sub-matrix defined by the subset of the rows of $U$, such that we select the row $\mathbf{d} \equiv (d_1, \ldots, d_{T/2})$ if it holds that $\forall j, d_j \leq d_{j+1}$. Note that there are exactly $N$ such rows, similarly to the number of columns, which can be intuitively understood since inside the imaginary bucket defining the columns there is no meaning of order in the balls, and having imposed the restriction $\forall j, d_j \leq d_{j+1}$ on the ${}^T\!/_2$ length tuple $\mathbf{d}$, there is no longer a degree of freedom to order the 'colors' in $\mathbf{d}$, reducing the number of rows from $M^{T/2}$ to $N$. Thus, in the resulting sub-matrix, denoted by $\bar{U} \in \mathbb{R}^{N \times N}$, not only do the columns correspond to the vectors of $\text{states}\left(\bar{R}, {}^T\!/_2\right)$, but also its rows, where the row specified by the tuple $\mathbf{d}$, corresponds to the vector $\mathbf{q}^{(T/2)} \in \text{states}\left(\bar{R}, {}^T\!/_2\right)$, such that for $r \in [\bar{R}] :\ q_r^{(T/2)} \equiv |\{j \in [{}^T\!/_2] | d_j = r\}|$ specifies the amount of repetitions of the number ('color') $r$ in the given tuple.

Accordingly, for each element of $\bar{U}$ the following holds:

$$\bar{U}_{\mathbf{q}^{(T/2)}, \mathbf{p}^{(T/2)}} = \prod_{r=1}^{\bar{R}} \prod_{j=1}^{T/2} Z_{rd_j}^{p_r^{(T/2)}}$$

$$\left(Z_{ij} = z^{\Omega^i \delta_{ij}}\right) \Rightarrow = z^{\sum_{j=1}^{T/2} \sum_{r=1}^{\bar{R}} p_r^{(T/2)} \Omega^r \delta_{rd_j}}$$

$$(\text{definition of } \delta_{ij}) \Rightarrow = z^{\sum_{j=1}^{T/2} \Omega^{d_j} p_{d_j}^{(T/2)}}$$

$$(\text{Grouping identical summands}) \Rightarrow = z^{\sum_{r=1}^{\bar{R}} \Omega^r \left|\{j \in [T/2] | d_j = r\}\right| p_r^{(T/2)}}$$

$$\left(q_r^{(T/2)} \equiv |\{j \in [T/2] | d_j = r\}|\right) \Rightarrow = z^{\sum_{r=1}^{\bar{R}} \Omega^r q_r^{(T/2)} p_r^{(T/2)}}$$

$$\begin{pmatrix} \bar{q}_r^{(T/2)} \equiv \Omega^{r/2} q_r^{(T/2)} \\ \bar{p}_r^{(T/2)} \equiv \Omega^{r/2} p_r^{(T/2)} \end{pmatrix} \Rightarrow = z^{\left\langle \bar{\mathbf{q}}^{(T/2)}, \bar{\mathbf{p}}^{(T/2)} \right\rangle}.$$

Since the elements of $\bar{U}$ are polynomial in $z$, then according to lemma 1, it is sufficient to show that there exists a single contributor to the determinant of $\bar{U}$ that has the highest degree of $z$ in order to ensure that the matrix is fully ranked for all values of $z$ but a finite set. Observing the summands of the determinant, i.e. $z^{\sum_{\mathbf{q}^{(T/2)} \in \text{states}(\bar{R}, T/2)} \left\langle \bar{\mathbf{q}}^{(T/2)}, \sigma(\bar{\mathbf{q}}^{(T/2)}) \right\rangle}$, where $\sigma$ is a permutation on the rows of $\bar{U}$, and noting that $\text{states}\left(\bar{R}, {}^T\!/_2\right)$ is a set of non-negative numbers by definition, lemma 2 assures us the existence of a strictly maximal contributor, satisfying the conditions of lemma 1.

We prove that the set $\left\{\mathbf{v}^{\mathbf{p}^{(T/2)}} \in \mathbb{R}^{M^{T/2}}\right\}_{\mathbf{p}^{(T/2)} \in \text{states}\left(\bar{R}, T/2\right)}$ is linearly independent by arranging it as the columns of the matrix $V \in \mathbb{R}^{M^{T/2} \times N}$, and showing that its rank equals to $N$. As in the case of $U$, we select the same sub-set of rows to form the sub-matrix $\bar{V} \in \mathbb{R}^{N \times N}$. We show that each of the diagonal elements of $\bar{V}$ is a polynomial function whose degree is strictly larger than the degree of all other elements in its row. As an immediate consequence, the product of the diagonal elements, i.e. $\prod_{i=1}^{N} \bar{V}_{ii}(z)$, has degree strictly larger than any other summand of the determinant $\det(\bar{V})$, and by employing lemma 1, $\bar{V}$ has full-rank for all values

of $z$ but a finite set. The degree of the polynomial function in each entry of $\bar{V}$ is given by:

$$\deg\left(\bar{V}_{\mathbf{d},\mathbf{p}^{(T/2)}}\right) = \max_{\substack{(\mathbf{p}^{(T/2-1)},\ldots,\mathbf{p}^{(1)}) \\ \in \text{trajectory}\left(\mathbf{p}^{(T/2)}\right)}} \deg\left(\prod_{r=1}^{\bar{R}} \prod_{j=T/2+1}^{T} Z_{r d_j}^{p_r^{(T-j+1)}}\right)$$

$$= \max_{\substack{(\mathbf{p}^{(T/2-1)},\ldots,\mathbf{p}^{(1)}) \\ \in \text{trajectory}\left(\mathbf{p}^{(T/2)}\right)}} \deg\left(z^{\sum_{j=T/2+1}^{T}\sum_{r=1}^{\bar{R}} \Omega^r p_r^{(T-j+1)} \delta_{r d_j}}\right)$$

$$= \max_{\substack{(\mathbf{p}^{(T/2-1)},\ldots,\mathbf{p}^{(1)}) \\ \in \text{trajectory}\left(\mathbf{p}^{(T/2)}\right)}} \sum_{j=T/2+1}^{T} \Omega^{d_j} p_{d_j}^{(T-j+1)}.$$

The above can be formulated as the following combinatorial optimization problem. We are given an initial state $\mathbf{p}^{(T/2)}$ of the bucket of $T/2$ balls of $\bar{R}$ colors and a sequence of colors $\mathbf{d} = (d_{T/2+1},\ldots,d_T)$. At time-step $j$ one ball is taken out of the bucket and yields a reward of $\Omega^{d_j} p_{d_j}^{(T-j+1)}$, i.e. the number of remaining balls of color $d_j$ times the weight $\Omega^{d_j}$. Finally, $\deg(\bar{V}_{\mathbf{d},\mathbf{p}^{(T/2)}})$ is the accumulated reward of the optimal strategy of emptying the bucket. In lemma 3 we prove that there exists a value of $\Omega$ such that for every sequence of colors $\mathbf{d}$, i.e. a row of $\bar{V}$, the maximal reward over all possible initial states is solely attained at the state $\mathbf{q}^{(T/2)}$ corresponding to $\mathbf{d}$, i.e. $q_r^{(T/2)} = |\{j \in \{T/2+1,\ldots,T\}|d_j = r\}|$. Hence, $\deg(\bar{V}_{ii})$ is indeed strictly larger than the degree of all other elements in the $i$'th row.

Having proved that both $U$ and $V$ have rank $N \equiv \left(\!\!\binom{\bar{R}}{T/2}\!\!\right)$ for all values of $z$ but a finite set, we know there exists a value of $z$ for which $\text{rank}(B) = N$, and the theorem follows.

□

### B.3.3 TECHNICAL LEMMAS AND CLAIMS

In this section we prove a series of useful technical lemmas, that we have employed in our proof for the case of deep RACs, as described in section B.3.2. We begin by quoting a claim regarding the prevalence of the maximal matrix rank for matrices whose entries are polynomial functions:

**Claim 5.** *Let $M, N, K \in \mathbb{N}$, $1 \leq r \leq \min\{M,N\}$ and a polynomial mapping $A : \mathbb{R}^K \to \mathbb{R}^{M \times N}$, i.e. for every $i \in [M]$ and $j \in [N]$ it holds that $A_{ij} : \mathbb{R}^K \to \mathbb{R}$ is a polynomial function. If there exists a point $\mathbf{x} \in \mathbb{R}^K$ s.t. $\text{rank}(A(\mathbf{x})) \geq r$, then the set $\{\mathbf{x} \in \mathbb{R}^K : \text{rank}(A(\mathbf{x})) < r\}$ has zero measure (w.r.t. the Lebesgue measure over $\mathbb{R}^K$).*

*Proof.* See Sharir et al. (2016). □

Claim 5 implies that it suffices to show a specific assignment of the recurrent network weights for which the corresponding grid tensor matricization achieves a certain rank, in order to show this is a lower bound on its rank for all configurations of the network weights but a set of Lebesgue measure zero. Essentially, this means that it is enough to provide a specific assignment that achieves the required bound in theorem 1 in order to prove the theorem. Next, we show that for a matrix with entries that are polynomials in $x$, if a single contributor to the determinant has the highest degree of $x$, then the matrix is fully ranked for all values of $x$ but a finite set:

**Lemma 1.** *Let $A \in \mathbb{R}^{N \times N}$ be a matrix whose entries are polynomials in $x \in \mathbb{R}$. In this case, its determinant may be written as $\det(A) = \sum_{\sigma \in S_N} sgn(\sigma) p_\sigma(x)$, where $S_N$ is the symmetric group on $N$ elements and $p_\sigma(x)$ are polynomials defined by $p_\sigma(x) \equiv \prod_{i=1}^{N} A_{i\sigma(i)}(x)$, $\forall \sigma \in S_n$. Additionally, let there exist $\bar{\sigma}$ such that $\deg(p_{\bar{\sigma}}(x)) > \deg(p_\sigma(x))$ $\forall \sigma \neq \bar{\sigma}$. Then, for all values of $x$ but a finite set, $A$ is fully ranked.*

*Proof.* We show that in this case $\det(A)$, which is a polynomial in $x$ by its definition, is not the zero polynomial. Accordingly, $\det(A) \neq 0$ for all values of $x$ but a finite set. Denoting $t \equiv \deg(p_{\bar{\sigma}}(x))$, since $t > \deg(p_\sigma(x))$ $\forall \sigma \neq \bar{\sigma}$, a monomial of the form $c \cdot x^t, c \in \mathbb{R} \setminus \{0\}$ exists in $p_{\bar{\sigma}}(x)$ and doesn't exist in any $p_\sigma(x), \sigma \neq \bar{\sigma}$. This implies that $\det(A)$ is not the zero polynomial, since its leading term has a non-vanishing coefficient $sgn(\bar{\sigma}) \cdot c \neq 0$, and the lemma follows from the basic identity: $\det(A) \neq 0 \iff A$ is fully ranked. □

The above lemma assisted us in confirming that the assignment provided for the recurrent network weights indeed achieves the required grid tensor matricization rank of $\left(\!\!\binom{\bar{R}}{T/2}\!\!\right)$. The following lemma, establishes a useful relation we refer to as the *vector rearrangement inequality*:

**Lemma 2.** *Let $\{\mathbf{v}^{(i)}\}_{i=1}^N$ be a set of $N$ different vectors in $\mathbb{R}^{\bar{R}}$ such that $\forall i \in [N]$, $j \in [\bar{R}] : v_j^{(i)} \geq 0$. Then, for all $\sigma \in S_N$ such that $\sigma \neq \mathbb{I}_N$, where $S_N$ is the symmetric group on $N$, it holds that:*

$$\sum_{i=1}^N \left\langle \mathbf{v}^{(i)}, \mathbf{v}^{(\sigma(i))} \right\rangle < \sum_{i=1}^N \left\| \mathbf{v}^{(i)} \right\|^2.$$

*Proof.* We rely on theorem 368 in Hardy et al. (1952), which implies that for a set of non-negative numbers $\{a^{(1)}, \ldots, a^{(N)}\}$ the following holds for all $\sigma \in S_N$:

$$\sum_{i=1}^N a^{(i)} a^{(\sigma(i))} \leq \sum_{i=1}^N (a^{(i)})^2, \tag{14}$$

with equality obtained only for $\sigma$ which upholds $\sigma(i) = j \iff a^{(i)} = a^{(j)}$. The above relation, referred to as the *rearrangement inequality*, holds separately for each component $j \in [\bar{R}]$ of the given vectors:

$$\sum_{i=1}^N v_j^{(i)} v_j^{(\sigma(i))} \leq \sum_{i=1}^N (v_j^{(i)})^2.$$

We now prove that for all $\sigma \in S_N$ such that $\sigma \neq \mathbb{I}_N$, $\exists \hat{j} \in [\bar{R}]$ for which the above inequality is hard, *i.e.*:

$$\sum_{i=1}^N v_{\hat{j}}^{(i)} v_{\hat{j}}^{(\sigma(i))} < \sum_{i=1}^N (v_{\hat{j}}^{(i)})^2. \tag{15}$$

By contradiction, assume that $\exists \hat{\sigma} \neq \mathbb{I}_N$ for which $\forall j \in [\bar{R}]$:

$$\sum_{i=1}^N v_j^{(i)} v_j^{(\hat{\sigma}(i))} = \sum_{i=1}^N (v_j^{(i)})^2.$$

From the conditions of achieving equality in the rearrangement inequality defined in eq. 14, it holds that $\forall j \in [\bar{R}] : v_j^{(\hat{\sigma}(i))} = v_j^{(i)}$, trivially entailing: $\mathbf{v}^{(\hat{\sigma}(i))} = \mathbf{v}^{(i)}$. Thus, $\hat{\sigma} \neq \mathbb{I}_N$ would yield a contradiction to $\{\mathbf{v}^{(i)}\}_{i=1}^N$ being a set of $N$ different vectors in $\mathbb{R}^{\bar{R}}$. Finally, the hard inequality of the lemma for $\sigma \neq \mathbb{I}_N$ is implied from eq. 15:

$$\sum_{i=1}^N \left\langle \mathbf{v}^{(i)}, \mathbf{v}^{(\sigma(i))} \right\rangle \equiv \sum_{i=1}^N \left( \sum_{j=1}^{\bar{R}} v_j^{(i)} v_j^{(\sigma(i))} \right) = \sum_{j=1}^{\bar{R}} \left( \sum_{i=1}^N v_j^{(i)} v_j^{(\sigma(i))} \right) < \sum_{j=1}^{\bar{R}} \left( \sum_{i=1}^N (v_j^{(i)})^2 \right) = \sum_{i=1}^N \left\| \mathbf{v}^{(i)} \right\|^2.$$

$\square$

The vector rearrangement inequality in lemma 2, helped us ensure that our matrix of interest denoted $\bar{U}$ upholds the conditions of lemma 1 and is thus fully ranked. Below, we show an identity that allowed us to make combinatoric sense of a convoluted expression:

**Claim 6.** *Let $\bar{R}$ and $M$ be positive integers, let $Z \in \mathbb{R}^{\bar{R} \times M}$ be a matrix, and let $\mathcal{A}$ be a tensor with $T$ modes, each of dimension $M$, defined by $\mathcal{A}_{d_1, \ldots, d_T} \equiv \prod_{t=T/2+1}^T \sum_{r=1}^{\bar{R}} \prod_{j=1}^t Z_{rd_j}$, where $d_1, \ldots, d_T \in [M]$. Then, the following identity holds:*

$$\mathcal{A}_{d_1, \ldots, d_T} = \sum_{\substack{\mathbf{p}^{(T/2)} \\ \in \text{states}(\bar{R}, T/2)}} \sum_{\substack{(\mathbf{p}^{(T/2-1)}, \ldots, \mathbf{p}^{(1)}) \\ \in \text{trajectory}(\mathbf{p}^{(T/2)})}} \prod_{r=1}^{\bar{R}} \left( \prod_{j=1}^{T/2} Z_{rd_j}^{p_r^{(T/2)}} \right) \left( \prod_{j=T/2+1}^T Z_{rd_j}^{p_r^{(T-j+1)}} \right),$$

*where* $\text{states}(\bar{R}, K) \equiv \{\mathbf{p}^{(K)} \in (\mathbb{N} \cup \{0\})^{\bar{R}} | \sum_{i=1}^{\bar{R}} p_i = K\}$, *and* $\text{trajectory}(\mathbf{p}^{(K)}) \equiv \{(\mathbf{p}^{(K-1)}, \ldots, \mathbf{p}^{(1)}) | \forall k \in [K-1], (\mathbf{p}^{(k)} \in \text{states}(\bar{R}, k) \wedge \forall r \in [\bar{R}], p_r^{(k)} \leq p_r^{(k+1)})\}$. [8]

*Proof.* We will prove the following more general identity by induction. For any $k \in [T]$, define $\mathcal{A}_{d_1, \ldots, d_T}^{(k)} \equiv \prod_{t=k}^T \sum_{r=1}^{\bar{R}} \prod_{j=1}^t Z_{rd_j}$, then the following identity holds:

$$\mathcal{A}_{d_1, \ldots, d_T}^{(k)} = \sum_{\substack{\mathbf{p}^{(T-k+1)} \\ \in \text{states}(\bar{R}, T-k+1)}} \sum_{\substack{(\mathbf{p}^{(T-k)}, \ldots, \mathbf{p}^{(1)}) \\ \in \text{trajectory}(\mathbf{p}^{(T-k+1)})}} \prod_{r=1}^{\bar{R}} \left( \prod_{j=1}^{k-1} Z_{rd_j}^{p_r^{(T-k+1)}} \right) \left( \prod_{j=k}^T Z_{rd_j}^{p_r^{(T-j+1)}} \right).$$

---

[8] See section B.3.2 for a more intuitive definition of the sets $\text{states}(\bar{R}, K)$ and $\text{trajectory}(\mathbf{p}^{(T-k+1)})$.

The above identity coincides with our claim for $k = T/2 + 1$ We begin with the base case of $k = T$, for which the set $\text{states}\left(\bar{R}, 1\right)$ simply equals to the unit vectors of $(\mathbb{N} \cup \{0\})^{\bar{R}}$, i.e. for each such $\mathbf{p}^{(1)}$ there exists $\bar{r} \in [\bar{R}]$ such that $p_r^{(1)} = \delta_{\bar{r}r} \equiv \begin{cases} 1 & \bar{r} = r \\ 0 & \bar{r} \neq r \end{cases}$. Thus, the following equalities hold:

$$\sum_{\mathbf{p}^{(1)} \in \text{states}\left(\bar{R}, 1\right)} \prod_{r=1}^{\bar{R}} \prod_{j=1}^{T} Z_{rd_j}^{p_r^{(1)}} = \sum_{\bar{r}=1}^{\bar{R}} \prod_{r=1}^{\bar{R}} \prod_{j=1}^{T} Z_{rd_j}^{\delta_{\bar{r}r}} = \sum_{\bar{r}=1}^{\bar{R}} \prod_{j=1}^{T} Z_{\bar{r}d_j} = \mathcal{A}_{d_1,\ldots,d_T}^{(T)}.$$

By induction on $k$, we assume that the claim holds for $\mathcal{A}^{(k+1)}$ and prove it on $\mathcal{A}^{(k)}$. First notice that we can rewrite our claim for $k < T$ as:

$$\mathcal{A}_{d_1,\ldots,d_T}^{(k)} = \sum_{\substack{\mathbf{p}^{(T-k+1)} \\ \in \text{states}\left(\bar{R}, T-k+1\right)}} \sum_{\substack{(\mathbf{p}^{(T-k)},\ldots,\mathbf{p}^{(1)}) \\ \in \text{trajectory}\left(\mathbf{p}^{(T-k+1)}\right)}} \prod_{r=1}^{\bar{R}} \left(\prod_{j=1}^{k} Z_{rd_j}^{p_r^{(T-k+1)}}\right) \left(\prod_{j=k+1}^{T} Z_{rd_j}^{p_r^{(T-j+1)}}\right), \tag{16}$$

where we simply moved the $k$'th term $Z_{rd_k}^{p_r^{(k)}}$ in the right product expression to the left product. We can also can rewrite $\mathcal{A}^{(k)}$ as a recursive formula:

$$\mathcal{A}_{d_1,\ldots,d_T}^{(k)} = \left(\sum_{r=1}^{\bar{R}} \prod_{j=1}^{k} Z_{rd_j}\right) \cdot \mathcal{A}_{d_1,\ldots,d_T}^{(k+1)} = \left(\sum_{\bar{r}=1}^{\bar{R}} \prod_{r=1}^{\bar{R}} \prod_{j=1}^{k} Z_{rd_j}^{\delta_{\bar{r}r}}\right) \cdot \mathcal{A}_{d_1,\ldots,d_T}^{(k+1)}$$

. Then, employing our induction assumption for $\mathcal{A}^{(k+1)}$, results in:

$$\mathcal{A}_{d_1,\ldots,d_T}^{(k)} = \left(\sum_{\bar{r}=1}^{\bar{R}} \prod_{r=1}^{\bar{R}} \prod_{j=1}^{k} Z_{rd_j}^{\delta_{\bar{r}r}}\right) \sum_{\substack{\mathbf{p}^{(T-k)} \\ \in \text{states}\left(\bar{R}, T-k\right)}} \sum_{\substack{(\mathbf{p}^{(T-k-1)},\ldots,\mathbf{p}^{(1)}) \\ \in \text{trajectory}\left(\mathbf{p}^{(T-k)}\right)}} \prod_{r=1}^{\bar{R}} \left(\prod_{j=1}^{k} Z_{rd_j}^{p_r^{(T-k)}}\right) \left(\prod_{j=k+1}^{T} Z_{rd_j}^{p_r^{(T-j+1)}}\right)$$

$$= \sum_{\bar{r}=1}^{\bar{R}} \sum_{\substack{\mathbf{p}^{(T-k)} \\ \in \text{states}\left(\bar{R}, T-k\right)}} \sum_{\substack{(\mathbf{p}^{(T-k-1)},\ldots,\mathbf{p}^{(1)}) \\ \in \text{trajectory}\left(\mathbf{p}^{(T-k)}\right)}} \prod_{r=1}^{\bar{R}} \left(\prod_{j=1}^{k} Z_{rd_j}^{p_r^{(T-k)}+\delta_{\bar{r}r}}\right) \left(\prod_{j=k+1}^{T} Z_{rd_j}^{p_r^{(T-j+1)}}\right) \tag{17}$$

To prove that the right hand side of eq. 17 is equal to our alternative form of our claim given by eq. 16, it is sufficient to show a bijective mapping from the terms in the sum of eq. 17, each specified by a sequence $(\bar{r}, \mathbf{p}^{(T-k)}, \ldots, \mathbf{p}^{(1)})$, where $\bar{r} \in [\bar{R}]$, $\mathbf{p}^{(T-k)} \in \text{states}\left(\bar{R}, T-k\right)$, and $(\mathbf{p}^{(T-k-1)}, \ldots, \mathbf{p}^{(1)}) \in \text{trajectory}\left(\mathbf{p}^{(T-k)}\right)$, to the terms in the sum of eq. 16, each specified by a similar sequence $(\mathbf{p}^{(T-k+1)}, \mathbf{p}^{(T-k)}, \ldots, \mathbf{p}^{(1)})$, where $\mathbf{p}^{(T-k+1)} \in \text{states}\left(\bar{R}, T-k+1\right)$ and $(\mathbf{p}^{(T-k)}, \ldots, \mathbf{p}^{(1)}) \in \text{trajectory}\left(\mathbf{p}^{(T-k+1)}\right)$.

Let $\phi$ be a mapping such that $(\bar{r}, \mathbf{p}^{(T-k)}, \ldots, \mathbf{p}^{(1)}) \overset{\phi}{\mapsto} (\mathbf{p}^{(T-k+1)}, \mathbf{p}^{(T-k)}, \ldots, \mathbf{p}^{(1)})$, where $p_r^{(T-k+1)} \equiv p_r^{(T-k)} + \delta_{\bar{r}r}$. $\phi$ is injective, because if $\phi(\bar{r}_1, \mathbf{p}^{(T-k,1)}, \ldots, \mathbf{p}^{(1,1)}) = \phi(\bar{r}_2, \mathbf{p}^{(T-k,2)}, \ldots, \mathbf{p}^{(1,2)})$ then for all $j \in \{1, \ldots, T-k+1\}$ it holds that $\mathbf{p}^{(j,1)} = \mathbf{p}^{(j,2)}$, and specifically for $\mathbf{p}^{(T-k+1,1)} = \mathbf{p}^{(T-k+1,2)}$ it entails that $\delta_{\bar{r}_1 r} = \delta_{\bar{r}_2 r}$, and thus $\bar{r}_1 = \bar{r}_2$. $\phi$ is surjective, because for any sequence $(\mathbf{p}^{(T-k+1)}, \mathbf{p}^{(T-k)}, \ldots, \mathbf{p}^{(1)})$, for which it holds that $\forall j, \mathbf{p}^{(j)} \in (\mathbb{N} \cup \{0\})^{\bar{R}}$, $\sum_{r=1}^{\bar{R}} p_r^{(j)} = j$, and $\forall r, p_r^{(j)} \leq p_r^{(j+1)}$, then it must also holds that $p_r^{(T-k+1)} - p_r^{(T-k)} = \delta_{\bar{r}r}$ for some $\bar{r}$, since $\sum_{r=1}^{\bar{R}} (p_r^{(T-k+1)} - p_r^{(T-k)}) = (T-k+1) - (T-k) = 1$ and every summand is a non-negative integer.

$\square$

Finally, lemma 3 assists us in ensuring that our matrix of interest denoted $\bar{V}$ upholds the conditions of lemma 1 and is thus fully ranked:

**Lemma 3.** *Let $\Omega \in \mathbb{R}_+$ be a positive real number. For every $\mathbf{p}^{(T/2)} \in \text{states}\left(\bar{R}, T/2\right)$ (see definition in claim 6) and every $\mathbf{d} = (d_{T/2+1}, \ldots, d_T) \in [\bar{R}]^{T/2}$, where $\forall j, d_j \leq d_{j+1}$, we define the following optimization problem:*

$$f(\mathbf{d}, \mathbf{p}^{(T/2)}) = \max_{\substack{(\mathbf{p}^{(T/2-1)}, \ldots, \mathbf{p}^{(1)}) \\ \in \text{trajectory}\left(\mathbf{p}^{(T/2)}\right)}} \sum_{j=T/2+1}^{T} \Omega^{d_j} p_{d_j}^{(T-j+1)},$$

*where* trajectory $\left(\mathbf{p}^{(T/2)}\right)$ *is defined as in claim 6. Then, there exists* $\Omega$ *such that for every such* $\mathbf{d}$ *the maximal value of* $f(\mathbf{d}, \mathbf{p}^{(T/2)})$ *over all* $\mathbf{p}^{(T/2)} \in$ *states* $\left(\bar{R}, T/2\right)$ *is strictly attained at* $\hat{\mathbf{p}}^{(T/2)}$ *defined by* $\hat{p}_r^{(T/2)} = |\{j \in \{T/2 + 1, \ldots, T\} | d_j = r\}|$.

*Proof.* We will prove the lemma by first considering a simple strategy for choosing the trajectory for the case of $f(\mathbf{d}, \hat{\mathbf{p}}^{(T/2)})$, achieving a certain reward $\rho^*$, and then showing that it is strictly larger than the rewards attained for all of the possible trajectories of any other $\mathbf{p}^{(T/2)} \neq \hat{\mathbf{p}}^{(T/2)}$.

Our basic strategy is to always pick the ball of the lowest available color $r$. More specifically, if $\hat{p}_1^{(T/2)} > 0$, then in the first $\hat{p}_1^{(T/2)}$ time-steps we remove balls of the color 1, in the process of which we accept a reward of $\Omega^1 \hat{p}_1^{(T/2)}$ in the first time-step, $\Omega^1(\hat{p}_1^{(T/2)} - 1)$ in the second time-step, and so on to a total reward of $\Omega^1 \sum_{i=1}^{\hat{p}_1^{(T/2)}} i$. Then, we proceed to removing $\hat{p}_2^{(T/2)}$ balls of color 2, and so forth. This strategy will result in an accumulated reward of:

$$\rho^* \equiv \sum_{r=1}^{\bar{R}} \Omega^r \sum_{i=1}^{\hat{p}_r^{(T/2)}} i.$$

Next, we assume by contradiction that there exists $\mathbf{p}^{(T/2)} \neq \hat{\mathbf{p}}^{(T/2)}$ such that $\rho \equiv f(\mathbf{d}, \mathbf{p}^{(T/2)}) \geq \rho^*$. We show by induction that this implies $\forall r, p_r^{(T/2)} \geq \hat{p}_r^{(T/2)}$, which would result in a contradiction, since per our assumption $\mathbf{p}^{(T/2)} \neq \hat{\mathbf{p}}^{(T/2)}$ this means that there is $r$ such that $p_r^{(T/2)} > \hat{p}_r^{(T/2)}$, but since $\mathbf{p}^{(T/2)}, \hat{\mathbf{p}}^{(T/2)} \in$ states $\left(\bar{R}, T/2\right)$ then the following contradiction arises $T/2 = \sum_{r=1}^{\bar{R}} p_r^{(T/2)} > \sum_{r=1}^{\bar{R}} \hat{p}_r^{(T/2)} = T/2$. More specifically, we show that our assumption entails that for all $r$ starting with $r = \bar{R}$ and down to $r = 1$, it holds that $p_r^{(T/2)} \geq \hat{p}_r^{(T/2)}$.

Before we begin proving the induction, we choose a value for $\Omega$ that upholds $\Omega > (T/2)^2$ such that the following condition holds: for any $r \in [\bar{R}]$, the corresponding weight for the color $r$, i.e. $\Omega^r$, is strictly greater than $\Omega^{r-1}(T/2)^2$. Thus, adding the reward of even a single ball of color $r$ is always preferable over any possible amount of balls of color $r' < r$.

We begin with the base case of $r = \bar{R}$. If $\hat{p}^{(T/2)} = 0$ the claim is trivially satisfied. Otherwise, we assume by contradiction that $p_{\bar{R}}^{(T/2)} < \hat{p}_{\bar{R}}^{(T/2)}$. If $p_{\bar{R}}^{(T/2)} = 0$, then the weight of the color $\bar{R}$ is not part of the total reward $\rho$, and per our choice of $\Omega$ it must hold that $\rho < \rho^*$ since $\rho^*$ does include a term of $\Omega^{\bar{R}}$ by definition. Now, we examine the last state of the trajectory $\mathbf{p}^{(1)}$, where there is a single ball left in the bucket. Per our choice of $\Omega$, if $p_{\bar{R}}^{(1)} = 0$, then once again $\rho < \rho^*$, implying that $p_{\bar{R}}^{(1)} = 1$. Following the same logic, for $j \in [p_{\bar{R}}^{(T/2)}]$, it holds that $p_{\bar{R}}^{(j)} = j$. Thus the total contribution of the $\bar{R}$'th weight is at most:

$$\Omega^{\bar{R}} \left( (\hat{p}_{\bar{R}}^{(T/2)} - p_{\bar{R}}^{(T/2)}) \cdot p_{\bar{R}}^{(T/2)} + \sum_{i=1}^{p_{\bar{R}}^{(T/2)}} i \right). \tag{18}$$

This is because before spending all of the $p_{\bar{R}}^{(T/2)}$ balls of color $\bar{R}$ at the end, there are another $(\hat{p}_{\bar{R}}^{(T/2)} - p_{\bar{R}}^{(T/2)})$ time-steps at which we add to the reward a value of $p_{\bar{R}}^{(T/2)}$. However, since eq. 18 is strictly less than the corresponding contribution of $\Omega^{\bar{R}}$ in $\rho^*$: $\Omega^{\bar{R}} \sum_{i=1}^{\hat{p}_{\bar{R}}^{(T/2)}} i$, then it follows that $\rho < \rho^*$, in contradiction to our assumption, which implies that to uphold the assumption the following must hold: $p_{\bar{R}}^{(T/2)} \geq \hat{p}_{\bar{R}}^{(T/2)}$, proving the induction base.

Assuming our induction hypothesis holds for all $r' > r$, we show it also holds for $r$. Similar to our base case, if $\hat{p}_r^{(T/2)} = 0$ then our claim is trivially satisfied, and likewise if $p_r^{(T/2)} = 0$, hence it remains to show that the case of $p_r^{(T/2)} < \hat{p}_{\bar{R}}^{(T/2)}$ is not possible. First, according to our hypothesis, $\forall r' > r, p_{r'}^{(T/2)} \geq \hat{p}_{r'}^{(T/2)}$, and per our choice of $\Omega$, the contributions to the reward of all of the weights for $r' > r$, are at most $\sum_{r'=r+1}^{\bar{R}} \Omega^{r'} \sum_{i=1}^{\hat{p}_{r'}^{(T/2)}} i$, which is exactly equal to the corresponding contributions in $\rho^*$. This means that per our choice of $\Omega$ it suffices to show that the contributions originating in the color $r$ are strictly less than the ones in $\rho^*$ to prove our hypothesis. In this optimal setting, the state of the bucket at time-step $j = T/2 - \sum_{r'=r+1}^{\bar{R}} \hat{p}_{r'}^{(T/2)}$ must upholds $p_{r'}^{(j)} = \hat{p}_{r'}^{(T/2)}$ for $r' > r$, and zero otherwise. At this point, employing exactly the same logic as in our base case, the total contribution to the reward of the weight for the $r$'th color is at most:

$$\Omega^r \left( (\hat{p}_r^{(T/2)} - p_r^{(T/2)}) \cdot p_r^{(T/2)} + \sum_{i=1}^{p_r^{(T/2)}} i \right), \tag{19}$$

which is strictly less than the respective contribution in $\rho^*$.

$\square$

## C MATRICIZATION DEFINITION

Suppose $\mathcal{A} \in \mathbb{R}^{M \times \cdots \times M}$ is a tensor of order $T$, and let $(I, J)$ be a partition of $[T]$, *i.e.* $I$ and $J$ are disjoint subsets of $[T]$ whose union gives $[T]$. The *matricization of $\mathcal{A}$ w.r.t. the partition* $(I, J)$, denoted $[\![\mathcal{A}]\!]_{I,J}$, is the $M^{|I|}$-by-$M^{|J|}$ matrix holding the entries of $\mathcal{A}$ such that $\mathcal{A}_{d_1 \ldots d_T}$ is placed in row index $1 + \sum_{t=1}^{|I|}(d_{i_t} - 1)M^{|I|-t}$ and column index $1 + \sum_{t=1}^{|J|}(d_{j_t} - 1)M^{|J|-t}$.

