# OpenReview forum: "Benefits of Depth for Long-Term Memory of Recurrent Networks"
_ICLR.cc/2018/Conference — Invite to Workshop Track_

### Official Review · AnonReviewer2 · 2017-11-25
**An effect of increase of $L$ should be evaluated.**

**Rating:** 5
**Confidence:** 2

**Review:**

This paper investigates an effect of time dependencies in a specific type of RNN.

The idea is important and this paper seems sound. However, I am not sure that the main result (Theorem 1) explains an effect of depth sufficiently.

--Main comment
About the deep network case in Theorem 1, how $L$ affects the bound on ranks? In the current statement, the result seems independent to $L$ when $L \geq 2$. I think that this paper should quantify the effect of an increase of $L$.

--Sub comment
Numerical experiments for calculating the separation rank is necessary to provide evidence of the main result. Only a simple example will make this paper more convincing.

---

> ### Public Comment · (anonymous) · 2017-12-02
> **Response to Reviewer 2**
>
> We thank the reviewer for the time and feedback. Following is our response.
>
> Addressing the reviewer's main comment, indeed Theorem 1 does not include a separation between two deep recurrent networks of different depths - our analysis refers to the enhanced memory capacity of multiple layered recurrent networks versus single layered recurrent networks. Our treatment is analogous to a much wider and more mature line of work that deals with depth separation in feed-forward neural networks. In the feed-forward domain, various important and widely accepted papers give results which only separate between shallow and deep networks in order to show the advantages of depth in these networks (such as [1,2]). Formal theoretical literature which provides any such similar results on recurrent networks is scarce at best, and such questions have not been answered (or formally asked) to date for recurrent networks. Our focus on separating between shallow and deep networks is the natural starting point for this line of research; Theorem 1, which establishes a separation in the ability to integrate data throughout time between shallow and deep recurrent networks, constitutes a first of its kind theoretical assertion of superiority of deep recurrent networks.
>
> We wish to emphasize that even once the question is raised - "can the notion of depth enhanced long term-memory in recurrent networks be formalized?" and a mathematical infrastructure is set-up in the form of the Start-End separation rank with grid tensors rank bounding it, establishing Theorem 1 is highly non-trivial. As can be seen in the supplementary material, the proof involves a considerable "legwork" which integrates tools and results from various fields (measure theory, tensorial analysis, combinatorics, graph theory and quantum physics). Accordingly, given the importance and contribution of the result in Theorem 1, we found the suggested task of separating the memory capacity of two arbitrary deep recurrent networks subsidiary in terms of contribution to the message of this paper. We agree that a finer investigation separating two recurrent networks of arbitrary depth is of relevance - it is in fact a part of a follow-up work, indicated by this paper, which we are presently pursuing (described in the last paragraph of the discussion section).
>
> Regarding the sub-comment made by the reviewer, our theoretical results guarantee that for almost any setting of the recurrent network's weights, theorem 1 holds. We have performed several sanity checks, which agreed with our conclusions. Having proved the theorem, we did not feel the need to include such empirical validations. However, we will gladly do so if it helps clarify or convey any message. We would appreciate a clarification regarding what specific convincing experiments the reviewer had in mind.
>
> References
> ----------------
> [1] Olivier Delalleau and Yoshua Bengio. Shallow vs. deep sum-product networks. In Advances in Neural Information Processing Systems, pages 666–674, 2011.
> [2] Guido F. Montúfar, Razvan Pascanu, KyungHyun Cho, and Yoshua Bengio. On the number of linear regions of deep neural networks. In Advances in Neural Information Processing Systems 27: Annual Conference on Neural Information Processing Systems 2014.

---

> ### Author Response · Authors · 2018-01-05
> **Dependence on L is addressed**
>
> We encourage the reviewer to examine our separate official comment regarding the upload of a paper revision, which addresses the dependence on L of the bounds.

---

### Official Review · AnonReviewer4 · 2017-12-08
**Interesting theory, could benefit from some experiments**

**Rating:** 7
**Confidence:** 3

**Review:**

After reading the authors's rebuttal I increased my score from a 7 to a 6.  I do think the paper would benefit from experimental results, but agree with the authors that the theoretical results are non-trivial and interesting on their own merit.

------------------------
The paper presents a theoretical analysis of depth in RNNs (technically a variant called RACs) i.e. stacking RNNs on top of one another, so that h_t^l (i.e. hidden state at time t and layer l is a function of h_t^{l-1} and h_{t-1}^{l})

The work is inspired by previous results for feed forward nets and CNNs. However, what is unique to RNNs is their ability to model long term dependencies across time.

To analyze this specific property, the authors propose a concept called "start-end rank" that essentially models the richness of the dependency between two disjoint subsets of inputs. Specifically, let S = {1, . . . , T/2} and E === {T/2 + 1, . . . , T}. sep_{S,E}(y) models the dependence between these two sets of time points. Specifically sep_{S,E}(y) = K means there exists g_s^k and g_e^k for k=1...K such that y(x) = \sum_{k} g_s^k(x_S) g_e^k(x_E).

Therefore sep_{S,E}(y) is the rank of a particular matricization of y (with respect to the partition S,E). If sep_{S,E}=1 then it is rank 1 (and would correspond to independence if y(x) was a probability distribution). A higher rank would correspond to more dependence across time.

(Comment: I believe if I understood the above correctly, it would be easier to explain tensors/matricization first and then introduce separation rank, since I think it much makes it clearer to explain. Right now the authors explain separation rank first and then discuss tensors / matricization).

Using this concept, the authors prove that deep recurrent networks can express functions that have exponentially higher start/end ranks than shallow RNNs.

I overall like the paper's theoretical results, but I have the following complaints:

(1)  I have the same question as the other reviewer. Why is Theorem 1 not a function of L?  Do the papers that prove similar theorems about ConvNets able to handle general L? What makes this more challenging? I feel if comparing L=2 vs L=3 is hard, the authors should be more up front about that in the introduction/abstract.

(2) I think it would have been stronger if the authors would have provided some empirical results validating their claims.

---

> ### Public Comment · (anonymous) · 2017-12-08
> **Response to Reviewer 4**
>
> We thank the reviewer for his feedback and suggestion, which we will take into consideration. We address the reviewer's reservations below.
>
> 1) The lower bound presented in the paper for a deep network is tied to the minimal case of L=2, which is the technical reason for the lower bound in theorem 1 lacking a dependence on L. We do not claim it is unachievable to receive specific bounds on  L=3 etc. However, we deem a separation between higher L's subsidiary in terms of the simple and unprecedented message of this paper, which brings forth a formal explanation for superiority of the prevalent architectural choice of stacked recurrent networks. Regarding the line of work that deals with depth separation in feed-forward neural networks, certainly there are important papers for which the formal results only separate between L=1 and higher L, such as the ones mentioned in the reply to reviewer 2. A separation between two networks of higher L's is of interest, however our work is the first to raise and formulate the depth efficiency question in terms of recurrent networks, and thus we consider the clear message attained by the presented separation in Theorem 1 important.
>
> 2) As we emphasize in the paper, our analysis provides a theoretical framework for a phenomenon that is empirically well-established in the literature. Moreover, our main result in Theorem 1 specifically addresses the issue of enhanced long-term memory in deep recurrent networks, for which a comprehensive empirical study was presented by [1]. We refer to this study in the second paragraph of the introduction and in last paragraph of the conclusion. The work in [1] constitutes an empirical motivator for our work, which is positioned to provide a first of its kind theoretical perspective for the above findings. We found the methodological experimental indications in [1], strengthened by a variety of other empirical works which employ deep recurrent networks for demanding sequential tasks, sufficient. Thus, we did not include further experimental validations, which in our view were not required given the evidence in the literature cited in the paper. However, we would appreciate suggestions by the reviewer for experiments not present in the mentioned literature that are required for strengthening our presentation. Alternatively, we suggest to emphasize the existence of such supporting experiments in our introduction, so that their role in motivating our theoretical results is even clearer.
>
>
> References
> ----------------
> [1] Michiel Hermans and Benjamin Schrauwen. Training and analysing deep recurrent neural networks. In Advances in Neural Information Processing Systems, pages 190–198, 2013.

---

> ### Author Response · Authors · 2018-01-05
> **Response to Reviewer 4**
>
> We thank the reviewer for considering our response and for supporting the paper.

---

### Official Review · AnonReviewer3 · 2017-12-15

**Rating:** 6
**Confidence:** 3

**Review:**

The paper proposes to use the start-end rank to measure the long-term dependency in RNNs. It shows that deep RNN is signficantly better than shallow one in this metric.

The theory part seems to be technical enough and interesting, though I haven't checked all the details. The main concern with the paper is that I am not sure whether the RAC studied by the paper is realistic enough for practice. Certain gating in RNN is very useful but I don't know whether one can train any reasonable RNN with all multiplicative gates. The paper will be much stronger if it has some experiments along this line.

---

> ### Author Response · Authors · 2018-01-05
> **Response to Reviewer 3**
>
> We thank the reviewer for the time and feedback. Our response follows.
>
> RACs are brought forth in our paper as a theoretical platform to investigate more common RNNs. The depth related effects studied in this paper depend on the recurrent network's structure rather than on specific activations. Empirical support for the specific RAC activations can be found in [1], which we mention in the paper. There, Multiplicative Integration Networks are shown to outperform common RNNs with additive integration. In section 3.1 they investigate RNNs with only multiplicative gates (in our terms - RACs) and find they preform comparably to vanilla RNNs in [2]. Furthermore, reference [1] shows evidence that under multiplicative integration the effect of squashing nonlinearities is diminished as they mostly operate in their linear regime, as opposed to the additive case where they are heavily influential (Fig. 1 (c),(d)).
>
> Thus, there is a clear empirical validation addressing the reviewer's concern, that RACs can be trained to perform relevant sequential tasks. Our paper merely uses RACs as surrogates to common RNNs and does not propose to use them in practice - even though, as shown in empirical studies mentioned above, they can be used in practice.
>
> References
> ___________________________
> [1] Yuhuai Wu, Saizheng Zhang, Ying Zhang, Yoshua Bengio, and Ruslan R Salakhutdinov. On multiplicative integration with recurrent neural networks. In Advances in Neural Information Processing Systems, 2016.
> [2] David Krueger and Roland Memisevic. Regularizing rnns by stabilizing activations. arXiv:1511.08400, 2015.

---

### Author Response · Authors · 2018-01-05
**Uploaded revision with an added subsection**

In accordance with main comments raised by reviewers 2 and 4 we have uploaded a version of the paper that has a new subsection, enumerated 4.2. Our main result (theorem 1),  rigorously proves a lower bound on the Start-End separation rank of depth L=2 recurrent networks. This proved L=2 lower bound also trivially applies to all networks of depth L>2, and thus constitutes a first of its kind exponential separation in memory capacity between deep recurrent networks and shallow ones. In the added subsection, we present a quantitative conjecture by which a tighter, depth dependent, lower bound holds for recurrent networks of depth L>2. We formally motivate this conjecture by the Tensor Networks construction of deep RACs. We emphasize that the originally submitted version included the Tensor Networks construction of deep RACs (Appendices A1-A3), which yields the added conjecture in a straightforward manner, as described in a newly added appendix section A4. Beyond this, the paper kept its original form.

This addition, which meets central questions raised by the reviewers, outlines further insight that is achieved by our analysis regarding the dependence of the Start-End separation rank on depth, and poses further investigation of this avenue as an open problem. We believe that the presented novel approach for theoretical analysis of long term memory in recurrent networks, along with the solidly proved main results separating L=2 deep networks from L=1 shallow networks, constitutes an important contribution, which is well-supplemented by the formally motivated conjecture in the added section 4.2.

---

### Decision · Program_Chairs · 2018-01-29
**ICLR 2018 Conference Acceptance Decision**

**Decision:**

Invite to Workshop Track

**Comment:**

This paper attempts a theoretical treatment of the influence of depth in RNNs on their ability to capture dependencies in the data. All reviewers found the theoretical contribution of the paper interesting, and while there were problems raised regarding formalisation, they appear to have been adequately addressed in the revisions to the paper. The main concern in all three reviews surrounds the evaluation, and weakness thereof. The overarching point of contention seems to be that the theory relates to a particular formulation of RNNs (RAC), causing doubts that the results lift to other architectural variants which are used to obtain state-of-the-art results on tasks such as language modelling. It seems that the paper could be significantly improved by the provision of stronger empirical results to support the theory, or a more convincing argument as to why the results should transfer from, say, RAC to LSTMs. The authors point to two papers on the matter in their response, but it is not clear this is a substitute for experimental validation. I find the paper a bit borderline because of this, and recommend redirection to the workshop.